# Learning Kernelized Contextual Bandits in a Distributed and Asynchronous Environment

**Chuanhao Li**[1]    **Huazheng Wang**[2]    **Mengdi Wang**[3]    **Hongning Wang**[1]
[1]University of Virginia    [2]Oregon State University    [3]Princeton University
`{cl5ev,hw5x}@virginia.edu`
`huazheng.wang@oregonstate.edu`   `mengdiw@princeton.edu`

## Abstract

Despite the recent advances in communication-efficient distributed bandit learning, most existing solutions are restricted to parametric models, e.g., linear bandits and generalized linear bandits (GLB). In comparison, kernel bandits, which search for non-parametric functions in a reproducing kernel Hilbert space (RKHS), offer higher modeling capacity. But the only existing work in distributed kernel bandits adopts a synchronous communication protocol, which greatly limits its practical use (e.g., every synchronization step requires all clients to participate and wait for data exchange). In this paper, in order to improve the robustness against delays and unavailability of clients that are common in practice, we propose the first asynchronous solution based on approximated kernel regression for distributed kernel bandit learning. A set of effective treatments are developed to ensure approximation quality and communication efficiency. Rigorous theoretical analysis about the regret and communication cost is provided; and extensive empirical evaluations demonstrate the effectiveness of our solution.

## 1    Introduction

There are many application scenarios where an environment repeatedly provides a learner with a set of candidate actions to choose from, and possibly some side information (aka., context) (Li et al., 2010a;b; Durand et al., 2018); and the learner, whose goal is to maximize cumulative reward over time, can only observe the reward corresponding to the chosen action. This is often modeled as a bandit learning problem (Abbasi-Yadkori et al., 2011; Krause & Ong, 2011), which exemplifies the well-known exploitation-exploration dilemma (Auer, 2002). Various modeling assumptions have been made about the relation between the context for each action and its expected reward. Compared with parametric bandits, such as linear and generalized linear bandits (Abbasi-Yadkori et al., 2011; Filippi et al., 2010), kernel/Gaussian process bandits (Valko et al., 2013; Srinivas et al., 2009) offer greater flexibility as they find non-parametric functions lying in a RKHS. And thus they have become a powerful tool for optimizing black box functions based on noisy observations in various applications, such as recommender systems (Vanchinathan et al., 2014), mobile health (Tewari & Murphy, 2017), environment monitoring (Srinivas et al., 2009), automatic machine learning (Li et al., 2017), cyber-physical systems (Lizotte et al., 2007; Li et al., 2016), etc.

Motivated by the rapid growth in affordability and availability of hardware resources, e.g., computer clusters or IoT devices, there is increasing interest in distributing the learning tasks, which gives rise to the recent research efforts in distributed bandits (Wang et al., 2019; Huang et al., 2021; Li & Wang, 2022a;b; Li et al., 2022; He et al., 2022), where $N$ clients collaboratively maximize the overall cumulative rewards over time $T$. As communication bandwidth is the key bottleneck in many distributed applications (Huang et al., 2013), these studies emphasize communication efficiency, i.e., incur sub-linear communication cost with respect to time $T$, while attaining near-optimal regret. However, most of these works are restricted to simple parametric models, like linear bandits (Wang et al., 2019; Huang et al., 2021; Li & Wang, 2022a; He et al., 2022) or GLB (Li & Wang, 2022b). The only exception is Li et al. (2022), who proposed the first algorithm for distributed kernel bandit that has sub-linear communication cost. They achieved this via a Nyström embedding function (Nyström, 1930) shared among all the clients, such that the clients only need to transfer the embedded statistics for joint kernelized estimation. Nevertheless, in their algorithm, the update of

the Nyström embedding function, as well as the communication of the embedded statistics, relies on a synchronization round that requires participation of all the clients. As is widely recognized in distributed optimization (Low et al., 2012; Xie et al., 2019; Lian et al., 2018; Chen et al., 2020; Lim et al., 2020) and distributed bandit learning (Li & Wang, 2022a; He et al., 2022), this design is vulnerable to stragglers (i.e., slower clients) in the system, i.e., the update procedure of Li et al. (2022) is paused until the slowest client responds. Due to device heterogeneity and network unreliability, this situation unfortunately is common especially at the scale of hundreds of devices/clients. Thus, *asynchronous communication* is preferred, as the server can readily perform model update when communication from a client is received, which is more robust against stragglers.

The main bottleneck in addressing this limitation of Li et al. (2022) lies in computing Nyström approximation under *asynchronous communication*. Specifically, during synchronization step, their algorithm first samples a small set of representative data points (i.e., the dictionary) from all clients, and then lets each client project their local data to the subspace spanned by this dictionary and share statistics about the projected data with others. However, new challenges arise in both algorithmic design and theoretical analysis when extending their solution to asynchronous communication, since a 'fresh' re-sample from the data of all clients is no longer possible, and each client has a different copy of the dictionary due to the asynchronous communication with the server, such that their local data will be projected to different subspaces, and thus causes difficulty in joint kernel estimation.

In this paper, we address these challenges and propose the first asynchronous algorithm for distributed kernelized contextual bandits. Compared with prior works in distributed bandits, our algorithm simultaneously enjoys the modeling capacity of non-parametric models and the improved robustness against delays and unavailability of clients, making it suitable for a wider range of applications. To ensure the approximation quality and compactness of the constructed dictionary in asynchronous communications, we design an incremental update procedure tailored to our problem setting with a variant of Ridge leverage score (RLS) sampling. Compared with the sampling procedure in prior works (Li et al., 2022; Calandriello et al., 2020), this requires specialized treatments in analysis, since the quality of the current dictionary now relies on all previous asynchronous communications. Moreover, to enable joint kernel estimation, we perform transformations on the server side to convert statistics from different clients to a common subspace, which to the best of our knowledge is also new in bandit literature. We rigorously proved that the proposed algorithm incurs an $\tilde{O}(N^2\gamma_T^3)$ communication cost, matching that of Li et al. (2022), where $\gamma_T$ is the maximum information gain, while still attaining the optimal $O(\sqrt{T}\gamma_T)$ regret.

## 2 RELATED WORKS

There have been increasing research efforts in distributed bandit learning in recent years, i.e., multiple agents collaborate in pure exploration (Hillel et al., 2013; Tao et al., 2019; Du et al., 2021), or regret minimization (Wang et al., 2019; Li & Wang, 2022a;b). They mainly differ in the relations of learning problems solved by the agents (i.e., homogeneous vs., heterogeneous) and the type of communication network (i.e., peer-to-peer (P2P) vs., star-shaped). However, most of these works assume linear reward functions, and the clients communicate by transferring the $O(d^2)$ sufficient statistics. For example, Korda et al. (2016) considered a peer-to-peer (P2P) communication network and assumes that the clients form clusters, i.e., each cluster is associated with a unique bandit problem. Huang et al. (2021) considered a star-shaped communication network as in our paper, but their proposed phase-based elimination algorithm only works in the fixed arm set setting. The closest works to ours are Wang et al. (2019); Dubey & Pentland (2020); Li & Wang (2022a); He et al. (2022), which propose event-triggered communication protocols to obtain sub-linear communication cost over time for distributed linear bandits with a time-varying arm set. In particular, Li & Wang (2022a) first considered the asynchronous communication setting for distributed bandit learning. Though their proposed algorithm avoids global synchronization (Wang et al., 2019), it still involves download to inactive clients. He et al. (2022) further improved their algorithm design and analysis, such that only the active client in each round needs to participate in communication.

In comparison, distributed kernelized contextual bandits still remain under-explored. Prior work in this direction assumes a local communication setting (Dubey et al., 2020), where the agent immediately shares the new raw data point to its neighbors after each interaction, and thus the communication cost is still linear over time. A recent work by Li et al. (2022) addresses this issue by

letting clients communicate via statistics computed using a shared Nyström embedding function (Calandriello et al., 2019; 2020). However, though their proposed algorithm attains sub-linear communication cost over time, it relies on a global synchronization operation similar to that of Wang et al. (2019) to update the embedding function and share the embedded statistics. In comparison, our proposed method in this paper effectively addresses this issue using a novel asynchronous update procedure for the embedding function, making asynchronous kernel bandit learning possible.

## 3 PRELIMINARIES

### 3.1 PROBLEM FORMULATION

We consider a learning system consisting of (1) $N$ clients that directly interact with the environment by taking actions and receiving the corresponding rewards, and (2) a central server that coordinates the communication among the clients to facilitate their learning. The clients cannot directly communicate with each other, but only with the central server, i.e., a star-shaped communication network. At each time step $t \in [T]$, an arbitrary client $i_t \in [N]$ becomes active and chooses an arm $\mathbf{x}_t$ from a candidate set $\mathcal{A}_t \subseteq \mathbb{R}^d$, and then receives the corresponding reward feedback $y_t = f(\mathbf{x}_t) + \eta_t \in \mathbb{R}$. Note that $\mathcal{A}_t$ is time-varying and assumed to be chosen by an oblivious adversary, $f$ denotes the unknown reward function shared by all clients, and $\eta_t$ denotes the noise. Moreover, under the asynchronous communication scheme considered in this paper, only the active client $i_t$ is allowed to communicate with the server, e.g., to send or receive updates, after its interaction at time step $t$.

**Kernelized Reward Function** Following Valko et al. (2013), we assume the unknown reward function $f$ lies in the RKHS, denoted as $\mathcal{H}$, such that the reward can be equivalently written as $y_t = \theta_\star^\top \phi(\mathbf{x}_t) + \eta_t$, where $\theta_\star \in \mathcal{H}$ is an unknown parameter vector and $\phi : \mathbb{R}^d \to \mathcal{H}$ is a known feature map associated with $\mathcal{H}$. We assume that $\eta_t$ is zero-mean $R$-sub-Gaussian conditioned on $\sigma\big((i_s, \mathbf{x}_s, \eta_s)_{s \in [t-1]}, i_t, \mathbf{x}_t\big)$, i.e., the $\sigma$-algebra generated by previous clients, their pulled arms, and the corresponding noises. In addition, there exists a positive definite kernel $k(\cdot, \cdot)$ associated with $\mathcal{H}$, and we assume $\forall \mathbf{x} \in \mathcal{A} := \cup_{t \in [T]} \mathcal{A}_t$ that, $\|\mathbf{x}\|_k \leq L$ and $\|f\|_k \leq S$ for some $L, S > 0$.

**Regret and Communication Cost** The learning system's goal is to minimize the cumulative (pseudo) regret for all $N$ clients, i.e., $R_T = \sum_{t=1}^T r_t$, where $r_t = \max_{\mathbf{x} \in \mathcal{A}_t} \phi(\mathbf{x})^\top \theta_\star - \phi(\mathbf{x}_t)^\top \theta_\star$. Meanwhile, the system also needs to keep the communication cost $C_T$ low, which is measured by the *total number of scalars* being transferred across the system up to time $T$.

### 3.2 KERNEL RIDGE REGRESSION & NYSTRÖM APPROXIMATION

Throughout the paper, we use $\mathcal{D} \subseteq [T]$ to denote a set of time steps and $|\mathcal{D}|$ as its size. The design matrix and reward vector constructed using data collected at these time steps, i.e., $\{\mathbf{x}_s, y_s\}_{s \in \mathcal{D}}$, are denoted as $\mathbf{X}_\mathcal{D} = [\mathbf{x}_s]_{s \in \mathcal{D}}^\top \in \mathbb{R}^{|\mathcal{D}| \times d}$ and $\mathbf{y}_\mathcal{D} = [y_s]_{s \in \mathcal{D}}^\top \in \mathbb{R}^{|\mathcal{D}|}$. Applying feature map $\phi(\cdot)$ to each row of $\mathbf{X}_\mathcal{D}$, we have $\mathbf{\Phi}_\mathcal{D} \in \mathbb{R}^{|\mathcal{D}| \times p}$, where $p$ denotes the dimension of $\mathcal{H}$ and is possibly infinite.

**Kernel Ridge Regression** Since the reward function $f$ is linear in $\mathcal{H}$, one can construct the Ridge regression estimator for $\theta_\star$ as,

$$\hat{\theta} = (\mathbf{\Phi}_\mathcal{D}^\top \mathbf{\Phi}_\mathcal{D} + \lambda \mathbf{I})^{-1} \mathbf{\Phi}_\mathcal{D}^\top \mathbf{y}_\mathcal{D}$$

where $\lambda > 0$ is the regularization parameter. This gives us the following estimated mean reward and standard deviation in the primal form for any arm $\mathbf{x} \in \mathcal{A}$:

$$\hat{\mu}(\mathbf{x}) = \phi(\mathbf{x})^\top \left(\mathbf{\Phi}_\mathcal{D}^\top \mathbf{\Phi}_\mathcal{D} + \lambda \mathbf{I}\right)^{-1} \left(\mathbf{\Phi}_\mathcal{D}^\top \mathbf{y}_\mathcal{D}\right)$$

$$\hat{\sigma}(\mathbf{x}) = \sqrt{\phi(\mathbf{x})^\top \left(\mathbf{\Phi}_\mathcal{D}^\top \mathbf{\Phi}_\mathcal{D} + \lambda \mathbf{I}\right)^{-1} \phi(\mathbf{x})}.$$

Note that directly working with the possibly infinite-dimension $\hat{\theta} \in \mathbb{R}^p$ is impractical. Instead, using the kernel trick (Valko et al., 2013; Li et al., 2022), we can obtain an equivalent dual form that only involves entries of the kernel matrix:

$$\hat{\mu}(\mathbf{x}) = \mathbf{K}_\mathcal{D}(\mathbf{x})^\top \left(\mathbf{K}_{\mathcal{D}, \mathcal{D}} + \lambda \mathbf{I}\right)^{-1} \mathbf{y}_\mathcal{D}$$

$$\hat{\sigma}(\mathbf{x}) = \lambda^{-1/2} \sqrt{k(\mathbf{x}, \mathbf{x}) - \mathbf{K}_\mathcal{D}(\mathbf{x})^\top \left(\mathbf{K}_{\mathcal{D}, \mathcal{D}} + \lambda \mathbf{I}\right)^{-1} \mathbf{K}_\mathcal{D}(\mathbf{x})}$$

$$(1)$$

where $\mathbf{K}_{\mathcal{D}}(\mathbf{x}) = \boldsymbol{\Phi}_{\mathcal{D}}\phi(\mathbf{x}) = [k(\mathbf{x}_s, \mathbf{x})]_{s \in \mathcal{D}}^{\top} \in \mathbb{R}^{|\mathcal{D}|}$ and $\mathbf{K}_{\mathcal{D},\mathcal{D}} = \boldsymbol{\Phi}_{\mathcal{D}}^{\top}\boldsymbol{\Phi}_{\mathcal{D}} = [k(\mathbf{x}_s, \mathbf{x}_{s'})]_{s,s' \in \mathcal{D}} \in \mathbb{R}^{|\mathcal{D}| \times |\mathcal{D}|}$.

**Nyström Approximation** Though equation 1 avoids directly working in $\mathcal{H}$, it requires computing the inverse of $\mathbf{K}_{\mathcal{D},\mathcal{D}}$, which is expensive in terms of both computation cost (Calandriello et al., 2019), i.e., $O(T^3)$ as $|\mathcal{D}| = O(T)$, and communication cost (Li et al., 2022), i.e., $O(T)$ as $\{(\mathbf{x}_s, y_s)\}_{s \in \mathcal{D}}$ needs to be transferred across the clients. Therefore, Nyström method is used to approximate equation 1, so clients can share embedded statistics, which improves communication efficiency.

As Calandriello et al. (2020); Li et al. (2022), we project the original dataset $\mathcal{D}^1$ to the subspace defined by a small representative subset $\mathcal{S} \subseteq \mathcal{D}$, i.e., the dictionary, and the orthogonal projection matrix is defined as

$$\mathbf{P}_{\mathcal{S}} = \boldsymbol{\Phi}_{\mathcal{S}}^{\top}(\boldsymbol{\Phi}_{\mathcal{S}}\boldsymbol{\Phi}_{\mathcal{S}}^{\top})^{-1}\boldsymbol{\Phi}_{\mathcal{S}} = \boldsymbol{\Phi}_{\mathcal{S}}^{\top}\mathbf{K}_{\mathcal{S},\mathcal{S}}^{-1}\boldsymbol{\Phi}_{\mathcal{S}} \in \mathbb{R}^{p \times p}.$$

Taking eigen-decomposition of $\mathbf{K}_{\mathcal{S},\mathcal{S}} = \mathbf{U}\boldsymbol{\Lambda}\mathbf{U}^{\top} \in \mathbb{R}^{|\mathcal{S}| \times |\mathcal{S}|}$, we can rewrite the orthogonal projection as $\mathbf{P}_{\mathcal{S}} = \boldsymbol{\Phi}_{\mathcal{S}}^{\top}\mathbf{U}\boldsymbol{\Lambda}^{-1/2}\boldsymbol{\Lambda}^{-1/2}\mathbf{U}^{\top}\boldsymbol{\Phi}_{\mathcal{S}}$, and define the Nyström embedding function as

$$z(\mathbf{x}; \mathcal{S}) = \mathbf{P}_{\mathcal{S}}^{1/2}\phi(\mathbf{x}) = \boldsymbol{\Lambda}^{-1/2}\mathbf{U}^{\top}\boldsymbol{\Phi}_{\mathcal{S}}\phi(\mathbf{x}) = \mathbf{K}_{\mathcal{S},\mathcal{S}}^{-1/2}\mathbf{K}_{\mathcal{S}}(\mathbf{x}),$$

which maps the data point $\mathbf{x}$ from $\mathbb{R}^d$ to $\mathbb{R}^{|\mathcal{S}|}$. Therefore, we can approximate the Ridge regression estimator on dataset $\mathcal{D}$ as $\tilde{\theta} = (\mathbf{P}_{\mathcal{S}}\boldsymbol{\Phi}_{\mathcal{D}}^{\top}\boldsymbol{\Phi}_{\mathcal{D}}\mathbf{P}_{\mathcal{S}} + \lambda\mathbf{I})^{-1}(\mathbf{P}_{\mathcal{S}}\boldsymbol{\Phi}_{\mathcal{D}}^{\top}\mathbf{y}_{\mathcal{D}})$, and equation 1 as

$$
\begin{aligned}
\tilde{\mu}(\mathbf{x}) &= z(\mathbf{x}; \mathcal{S})^{\top}(\mathbf{Z}_{\mathcal{D};\mathcal{S}}^{\top}\mathbf{Z}_{\mathcal{D};\mathcal{S}} + \lambda\mathbf{I})^{-1}\mathbf{Z}_{\mathcal{D};\mathcal{S}}^{\top}\mathbf{y}_{\mathcal{D}} \\
\tilde{\sigma}(\mathbf{x}) &= \lambda^{-1/2}\sqrt{k(\mathbf{x}, \mathbf{x}) - z(\mathbf{x}; \mathcal{S})^{\top}\mathbf{Z}_{\mathcal{D};\mathcal{S}}^{\top}\mathbf{Z}_{\mathcal{D};\mathcal{S}}[\mathbf{Z}_{\mathcal{D};\mathcal{S}}^{\top}\mathbf{Z}_{\mathcal{D};\mathcal{S}} + \lambda\mathbf{I}]^{-1}z(\mathbf{x}|\mathcal{S})}
\end{aligned}
\tag{2}
$$

where $\mathbf{Z}_{\mathcal{D};\mathcal{S}} \in \mathbb{R}^{|\mathcal{D}| \times |\mathcal{S}|}$ is obtained by applying $z(\cdot; \mathcal{S})$ to each row of $\mathbf{X}_{\mathcal{D}}$, i.e., $\mathbf{Z}_{\mathcal{D};\mathcal{S}} = \boldsymbol{\Phi}_{\mathcal{D}}\mathbf{P}_{\mathcal{S}}^{1/2}$. Note that the computation of $\tilde{\mu}(\mathbf{x})$ and $\tilde{\sigma}(\mathbf{x})$ only requires the embedded statistics, i.e., matrix $\mathbf{Z}_{\mathcal{D};\mathcal{S}}^{\top}\mathbf{Z}_{\mathcal{D};\mathcal{S}} \in \mathbb{R}^{|\mathcal{S}| \times |\mathcal{S}|}$ and vector $\mathbf{Z}_{\mathcal{D};\mathcal{S}}^{\top}\mathbf{y}_{\mathcal{D}} \in \mathbb{R}^{|\mathcal{S}|}$, which makes joint kernelized estimation among $N$ clients much more efficient in communication compared with equation 1.

## 4 METHODOLOGY

In this section, we propose and analyze the first asynchronous algorithm for distributed kernelized contextual bandit problem that addresses the challenges mentioned in Section 1, and name the resulting algorithm Async-KernelUCB, with its description given in Algorithm 1.

### 4.1 ALGORITHM

We denote the embedded statistics used in the computation of equation 2 by $\tilde{\mathbf{A}}(\mathcal{D}; \mathcal{S}) := \mathbf{Z}_{\mathcal{D};\mathcal{S}}^{\top}\mathbf{Z}_{\mathcal{D};\mathcal{S}}$ and $\tilde{\mathbf{b}}(\mathcal{D}; \mathcal{S}) := \mathbf{Z}_{\mathcal{D};\mathcal{S}}^{\top}\mathbf{y}_{\mathcal{D}}$, to explicitly emphasize they are computed by projecting the data points from dataset $\mathcal{D}$ to the subspace spanned by dictionary $\mathcal{S}$. We denote the sequence of time steps corresponding to the interactions between client $i$ and the environment up to time $t$ as $\mathcal{N}_t(i) = \{1 \leq s \leq t : i_s = i\}$ for $t \in [T]$. Throughout the paper, we reserve $k$ as the index for communication, and use $t_k \in [T]$ to denote the time step when the $k$-th communication happens. Moreover, as each client has a different copy of the embedding function and embedded statistics due to asynchronous communication, we use $\underline{k}(i)$ to denote the index of client $i$'s latest communication with the server, up to the $k$-th one: if client $i$ triggers the $k$-th communication, then $\underline{k}(i) = k$.

**Arm Selection** At each round $t \in [T]$, client $i_t \in [N]$ selects arm $\mathbf{x}_t$ from the candidate set $\mathcal{A}_t$ by maximizing the following upper confidence bound (line 5)

$$\mathbf{x}_t = \arg\max_{\mathbf{x} \in \mathcal{A}_t} \tilde{\mu}_{\underline{k}(i_t)}(\mathbf{x}) + \alpha\tilde{\sigma}_{\underline{k}(i_t)}(\mathbf{x}) \tag{3}$$

where $\tilde{\mu}_{\underline{k}(i_t)}(\mathbf{x})$ and $\tilde{\sigma}_{\underline{k}(i_t)}(\mathbf{x})$ are approximated mean and standard deviation of arm $\mathbf{x}$'s reward, computed using statistics $\tilde{\mathbf{A}}(\mathcal{D}_{\underline{k}(i_t)}, \mathcal{S}_{\underline{k}(i_t)})$ and $\tilde{\mathbf{b}}(\mathcal{D}_{\underline{k}(i_t)}, \mathcal{S}_{\underline{k}(i_t)})$ that client $i_t$ received from the server during the $\underline{k}(i_t)$-th communication. Proper choice of $\alpha$ is given in Lemma 4.4.

---

[1]Throughout this paper, we will often use the set of indices $\mathcal{D}$ (or $\mathcal{S}$) to refer to the actual dataset $\{\mathbf{x}_s, y_s\}_{s \in \mathcal{D}}$ (or dictionary $\{\mathbf{x}_s, y_s\}_{s \in \mathcal{S}}$) for simplicity.

---

**Algorithm 1** Asynchronous KernelUCB (Async-KernelUCB)

---

1: **Input:** $\alpha$, $\bar{q}$, communication threshold $D > 0$, regularization parameter $\lambda > 0$, $\delta \in (0, 1)$ and kernel function $k(\cdot, \cdot)$.
2: **Initialize** approximated mean and variance $\tilde{\mu}_0(\mathbf{x}) = 0$, $\tilde{\sigma}_0(\mathbf{x}) = \lambda^{-1/2}\sqrt{k(\mathbf{x}, \mathbf{x})}$, dataset $\mathcal{D}_0 = \emptyset$, dictionary $\mathcal{S}_0 = \emptyset$, index of communication $k = 0$, and $\mathcal{N}_0(i) = \emptyset$ for each client $i \in [N]$
3: **for** $t = 1, 2, ..., T$ **do**
4:     Client $i_t \in [N]$ becomes active, and observes arm set $\mathcal{A}_t$
5:     [Client $i_t$] Choose arm $\mathbf{x}_t \in \mathcal{A}_t$ according to equation 3, and observe reward $y_t$
6:     // Set $\mathcal{N}_t(i_t) = \mathcal{N}_{t-1}(i_t) \cup \{t\}$, and $\mathcal{N}_t(i) = \mathcal{N}_{t-1}(i)$ for $i \neq i_t$
7:     **if** $\sum_{s \in \mathcal{N}_t(i_t) \setminus \mathcal{N}_{t_{\underline{k}(i_t)}}(i_t)} \tilde{\sigma}^2_{\underline{k}(i_t)}(\mathbf{x}_s) > D$ **then**
        // Denote $\Delta \mathcal{D}_k = \mathcal{N}_t(i_t) \setminus \mathcal{N}_{t_{\underline{k}(i_t)}}(i_t)$, and set $k = k + 1$
8:         [Server $\rightarrow$ Client $i_t$] Send $\{\mathbf{x}_s, y_s\}_{s \in \mathcal{S}_{k-1}}$, $\tilde{\mathbf{A}}(\mathcal{D}_{k-1}; \mathcal{S}_{k-1})$, $\tilde{\mathbf{b}}(\mathcal{D}_{k-1}; \mathcal{S}_{k-1})$ to client $i_t$
9:         [Client $i_t$] Select $\Delta \mathcal{S}_k \subseteq \Delta \mathcal{D}_k$ via RLS sampling with probability $\bar{q}\tilde{\sigma}^2_{k-1}(\cdot)$
        // Set $\mathcal{S}_k = \mathcal{S}_{k-1} \cup \Delta \mathcal{S}_k$
10:        [Client $i_t$] Compute $\tilde{\mathbf{A}}(\Delta \mathcal{D}_k; \mathcal{S}_k)$, $\tilde{\mathbf{b}}(\Delta \mathcal{D}_k; \mathcal{S}_k)$
11:        [Client $i_t \rightarrow$ Server] Send $\{\mathbf{x}_s, y_s\}_{s \in \Delta \mathcal{S}_k}$, $\tilde{\mathbf{A}}(\Delta \mathcal{D}_k; \mathcal{S}_k)$ and $\tilde{\mathbf{b}}(\Delta \mathcal{D}_k; \mathcal{S}_k)$ to server
        // Set $\mathcal{D}_k = \mathcal{D}_{k-1} \cup \Delta \mathcal{D}_k$
12:        [Server] Compute $\tilde{\mathbf{A}}(\mathcal{D}_k; \mathcal{S}_k)$, $\tilde{\mathbf{b}}(\mathcal{D}_k; \mathcal{S}_k)$ according to equation 5
13:        [Server $\rightarrow$ Client $i_t$] Send $\tilde{\mathbf{A}}(\mathcal{D}_k; \mathcal{S}_k)$, $\tilde{\mathbf{b}}(\mathcal{D}_k; \mathcal{S}_k)$ to client $i_t$
14:        [Client $i_t$] Update $\tilde{\mu}_k(\cdot)$ and $\tilde{\sigma}_k(\cdot)$ using $\tilde{\mathbf{A}}(\mathcal{D}_k; \mathcal{S}_k)$, $\tilde{\mathbf{b}}(\mathcal{D}_k; \mathcal{S}_k)$ according to equation 2
15:     **end if**
16: **end for**

---

**Event-triggered Asynchronous Communication** After the interaction at time step $t$, $\tilde{\mu}_{\underline{k}(i_t)}(\cdot)$ and $\tilde{\sigma}_{\underline{k}(i_t)}(\cdot)$ of active client $i_t$ will only be updated if the following event is true (line 7):

$$\sum_{s \in \mathcal{N}_t(i_t) \setminus \mathcal{N}_{t_{\underline{k}(i_t)}}(i_t)} \tilde{\sigma}^2_{\underline{k}(i_t)}(\mathbf{x}_s) > D, \tag{4}$$

where $D > 0$ denotes the communication threshold. This measures whether sufficient amount of new information has been collected by client $i_t$ since its lastest (the $\underline{k}(i_t)$-th) communication with the server. If true, communication between client $i_t$ and the server is triggered (line 8-14), where the update procedure described in the following paragraphs will be performed. And this procedure is also illustrated in Figure 1.

**Dictionary and Embedded Statistics Update** During the $k$-th communication, the server first sends its latest dictionary $\mathcal{S}_{k-1}$, as well as its latest embedded statistics $\tilde{\mathbf{A}}(\mathcal{D}_{k-1}; \mathcal{S}_{k-1})$ and $\tilde{\mathbf{b}}(\mathcal{D}_{k-1}; \mathcal{S}_{k-1})$, to client $i_t$ (line 8), which is illustrated as the blue lines in Figure 1. Then client $i_t$ selects a subset $\Delta \mathcal{S}_k$ from the data it has collected since its lastest communication (line 9), i.e., $\Delta \mathcal{D}_k$, which will be used to incrementally update dictionary $\mathcal{S}_{k-1}$. This is done by sampling $q_{k,s} \sim \mathcal{B}(\tilde{p}_{k,s})$ for each data point with time index $s \in \Delta \mathcal{D}_k$, where $\tilde{p}_{k,s} := \bar{q}\tilde{\sigma}^2_{k-1}(\mathbf{x}_s)$. This can be considered as a variant of Ridge leverage score (RLS) sampling (Calandriello et al., 2020; Li et al., 2022). It is worth noting that the only purpose of sending $\tilde{\mathbf{A}}(\mathcal{D}_{k-1}; \mathcal{S}_{k-1})$ and $\tilde{\mathbf{b}}(\mathcal{D}_{k-1}; \mathcal{S}_{k-1})$ is to enable RLS sampling with the latest $\tilde{\sigma}^2_{k-1}(\cdot)$. Otherwise, client $i_t$, whose lastest communication with the server can be long time ago, would include unnecessary data points into $\Delta \mathcal{S}_k$ due to its unawareness of server's current status. We will demonstrate in the proof of Lemma 4.3 that our design here is necessary to obtain a compact dictionary under asynchronous communication. With the dictionary updated, client $i_t$ computes the embeddings of its new local data, i.e., $\tilde{\mathbf{A}}(\Delta \mathcal{D}_k; \mathcal{S}_k)$ and $\tilde{\mathbf{b}}(\Delta \mathcal{D}_k; \mathcal{S}_k)$, and sends them, as well as $\Delta \mathcal{S}_k$, to the server (the yellow lines in Figure 1).

As shown in Figure 1, the server stores: 1) the last received embedded statistics from each client $i \in [N]$, i.e., $\tilde{\mathbf{A}}(\mathcal{N}_{t_{\underline{k}(i)}}(i); \mathcal{S}_{\underline{k}(i)}) \in \mathbb{R}^{|\mathcal{S}_{\underline{k}(i)}| \times |\mathcal{S}_{\underline{k}(i)}|}$ and $\tilde{\mathbf{b}}(\mathcal{N}_{t_{\underline{k}(i)}}(i); \mathcal{S}_{\underline{k}(i)}) \in \mathbb{R}^{|\mathcal{S}_{\underline{k}(i)}|}$; 2) their corresponding dictionary $\mathcal{S}_{\underline{k}(i)}$. As mentioned earlier, due to asynchronous communication, the statistics from different clients are based on different dictionaries, which means they have different dimensions and thus *cannot be directly aggregated* as in Li et al. (2022). We propose to transform the statistics from each client $i \in [N]$ using the latest dictionary $\mathcal{S}_k$. This is based on the fact that

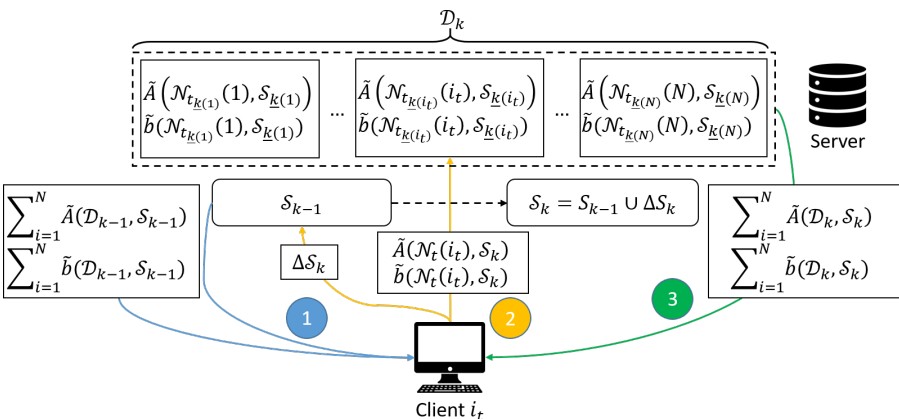

Figure 1: Illustration of asynchronous update of dictionary and embedded statistics

$\mathbf{Z}_{\mathcal{N}_{t_{\underline{k}(i)}}(i);\mathcal{S}_k} = \mathbf{\Phi}_{\mathcal{N}_{t_{\underline{k}(i)}}(i)}\mathbf{P}_{\mathcal{S}_k}^{1/2} = \mathbf{\Phi}_{\mathcal{N}_{t_{\underline{k}(i)}}(i)}\mathbf{P}_{\mathcal{S}_{\underline{k}(i)}}^{1/2}\mathbf{P}_{\mathcal{S}_{\underline{k}(i)}}^{-1/2}\mathbf{P}_{\mathcal{S}_k}^{1/2} = \mathbf{Z}_{\mathcal{N}_{t_{\underline{k}(i)}}(i);\mathcal{S}_{\underline{k}(i)}}\mathcal{T}_{\underline{k}(i),k}$, where the linear transformation $\mathcal{T}_{\underline{k}(i),k} := \mathbf{P}_{\mathcal{S}_{\underline{k}(i)}}^{-1/2}\mathbf{P}_{\mathcal{S}_k}^{1/2} = \mathbf{\Lambda}_{\mathcal{S}_{\underline{k}(i)}}^{1/2}\mathbf{U}_{\mathcal{S}_{\underline{k}(i)}}^{\top}\mathbf{\Phi}_{\mathcal{S}_{\underline{k}(i)}}\mathbf{\Phi}_{\mathcal{S}_k}^{\top}\mathbf{U}_{\mathcal{S}_k}\mathbf{\Lambda}_{\mathcal{S}_k}^{-1/2}$ serves the purpose. Hence, we have

$$\begin{aligned}
\tilde{\mathbf{A}}(\mathcal{N}_{t_{\underline{k}(i)}}(i);\mathcal{S}_k) &= \mathcal{T}_{\underline{k}(i),k}^{\top}\tilde{\mathbf{A}}(\mathcal{N}_{t_{\underline{k}(i)}}(i);\mathcal{S}_{\underline{k}(i)})\mathcal{T}_{\underline{k}(i),k}, \\
\tilde{b}(\mathcal{N}_{t_{\underline{k}(i)}}(i);\mathcal{S}_k) &= \mathcal{T}_{\underline{k}(i),k}^{\top}\tilde{b}(\mathcal{N}_{t_{\underline{k}(i)}}(i);\mathcal{S}_{\underline{k}(i)}),
\end{aligned} \tag{5}$$

which makes the statistics received from all clients have the same dimension $|\mathcal{S}_k|$. Then we compute $\tilde{A}(\mathcal{D}_k;\mathcal{S}_k) = \sum_{i=1}^{N}\tilde{\mathbf{A}}(\mathcal{N}_{t_{\underline{k}(i)}}(i);\mathcal{S}_k)$ and $\tilde{b}(\mathcal{D}_k;\mathcal{S}_k) = \sum_{i=1}^{N}\tilde{b}(\mathcal{N}_{t_{\underline{k}(i)}}(i);\mathcal{S}_k)$ (line 12), and send them to client $i_t$ to update its UCB (line 13-14), which is illustrated as the green line in Figure 1.

## 4.2 ANALYSIS OF DICTIONARY ACCURACY AND SIZE

As mentioned earlier, the key to low regret and low communication cost, is to have a dictionary $\mathcal{S}_k$ that can accurately approximate the dataset $\mathcal{D}_k$, while having a compact size $|\mathcal{S}_k|$. In this section, we show that this is possible with our update procedure in Section 4.1. First, we need some additional notations. We denote the total number of times up to time $T$ that communication is triggered, i.e., the number of times equation 4 is true, as $B$, where $B \in [0, T]$. Following Calandriello et al. (2020); Li et al. (2022), the approximation quality is formally defined using $\epsilon$-accuracy: if the event

$$\left\{(1-\epsilon)(\mathbf{\Phi}_{\mathcal{D}_k}^{\top}\mathbf{\Phi}_{\mathcal{D}_k} + \lambda\mathbf{I}) \preceq \mathbf{\Phi}_{\mathcal{D}_k}^{\top}\bar{\mathbf{S}}_k^{\top}\bar{\mathbf{S}}_k\mathbf{\Phi}_{\mathcal{D}_k} + \lambda\mathbf{I} \preceq (1+\epsilon)(\mathbf{\Phi}_{\mathcal{D}_k}^{\top}\mathbf{\Phi}_{\mathcal{D}_k} + \lambda\mathbf{I})\right\} \tag{6}$$

is true, then we say the dictionary $\mathcal{S}_k$ is $\epsilon$-accurate w.r.t. dataset $\mathcal{D}_k$, for some $\epsilon \in (0,1)$, where $\bar{\mathbf{S}}_k \in \mathbb{R}^{|\mathcal{D}_k| \times |\mathcal{D}_k|}$ denotes a diagonal matrix, with $s$-th diagonal entry equal to $q_{k,s}/\sqrt{\tilde{p}_{k,s}}$, where $q_{k,s} = 1$ if $s \in \mathcal{S}_k$, and $q_{k,s} = 0$, otherwise. Based on this notion, we prove Lemma 4.1 below.

**Lemma 4.1** (Dictionary Accuracy and Size). *With $\bar{q} = 4\ln(2\sqrt{2}T/\delta)\beta(1 + \epsilon/3)/\epsilon^2$, where $\beta := (1+\epsilon)/(1-\epsilon)$, and $\lambda \le k(\mathbf{x},\mathbf{x}), \forall\mathbf{x} \in \mathcal{A}$, we have with probability at least $1 - \delta$ that dictionary $\mathcal{S}_k$ is $\epsilon$-accurate w.r.t. dataset $\mathcal{D}_k$, and its size $|\mathcal{S}_k| \le 12\beta(1 + \beta D)\bar{q}\gamma_T, \forall k$, where $\delta \in (0,1)$.*

This shows that our incremental update procedure under asynchronous communication still matches the results in prior works that perform synchronous re-sampling over the whole dataset for dictionary update (Li et al., 2022; Calandriello et al., 2020). We provide a proof sketch for Lemma 4.1 below to highlight our technical novelty and provide the detailed proof in appendix.

*Proof Sketch of Lemma 4.1.* Let's define the unfavorable event $H_k = A_k \cup E_k$, where $A_k$ is the event that the dictionary $\mathcal{S}_k$ is not $\epsilon$-accurate w.r.t. $\mathcal{D}_k$, and $E_k$ is the event that the size of dictionary $|\mathcal{S}_k| > 12\beta(1 + \beta D)\bar{q}\gamma_T$. Therefore, the probability of $\cup_{k=0}^{B}H_k$ can be decomposed as

$$\mathbb{P}\big(\cup_{k=0}^{B}H_k\big) = \mathbb{P}\big(\cup_{k=0}^{B}A_k\big) + \mathbb{P}\big((\cup_{k=0}^{B}E_k) \cap (\cup_{k=0}^{B}A_k)^C\big).$$

*Bounding the first term:* In Calandriello et al. (2019; 2020); Li et al. (2022), the first term is further decomposed as $\mathbb{P}\big(\cup_{k=0}^{B}A_k\big) \le \sum_{k=1}^{B}\mathbf{P}(A_k \cap A_{k-1}^C)$, because dictionary $\mathcal{S}_k$ is constructed by a

fresh re-sampling over $\mathcal{D}_k$ using the latest approximated variance $\tilde{\sigma}^2_{k-1}(\cdot)$, and thus they only need to guarantee $\tilde{\sigma}^2_{k-1}(\cdot)$ is a good approximation to $\sigma^2_{k-1}(\cdot)$. In our case, $\mathcal{S}_k$ is incrementally updated in each communication, i.e., $\mathcal{S}_k = \cup_{k'=1}^k \Delta \mathcal{S}_{k'}$ where each $\Delta \mathcal{S}_{k'}$ is sampled using $\tilde{\sigma}^2_{k'-1}(\cdot)$. The accuracy of $\mathcal{S}_k$ depends on the accuracy of every $\mathcal{S}_{k'}$, i.e., $\cap_{k'=1}^{k-1} A_{k'}^C$. Therefore, we decompose $\mathbb{P}\left( \cup_{k=0}^B A_k \right) = 1 - \mathbb{P}\left( \cap_{k=0}^B A_k^C \right) = 1 - \prod_{k=1}^B [1 - \mathbb{P}\left( A_k | \cap_{k'=0}^{k-1} A_{k'}^C \right)] \leq \sum_{k=1}^B \mathbb{P}\left( A_k | \cap_{k'=0}^{k-1} A_{k'}^C \right)$ using Bayes theorem and Weierstrass product inequality, and bound each conditional probability separately, which leads to Lemma 4.2.

**Lemma 4.2** (Bounding $\sum_{k=1}^B \mathbb{P}\left( A_k | \cap_{k'=0}^{k-1} A_{k'}^C \right)$). *By setting $\bar{q} = 4 \ln(2\sqrt{2}T/\delta)\beta(1 + \epsilon/3)/\epsilon^2$, we have $\sum_{k=0}^B \mathbb{P}\left( A_k | \cap_{k'}^{k-1} A_{k'}^C \right) \leq \delta/2$, for $\delta \in (0, 1)$.*

*Bounding the second term:* The second term can be decomposed as $\mathbb{P}\left( (\cup_{k=0}^B E_k) \cap (\cup_{k=0}^B A_k)^C \right) \leq \sum_{k=0}^B \mathbb{P}\left( E_k \cap (\cap_{k=0}^B A_k^C) \right)$. Note that the size of dictionary $|\mathcal{S}_k| = \sum_{s \in \mathcal{D}_k} q_{k,s}$ by the definition of $q_{k,s}$, and its analysis relies on upper bounding $\sum_{s \in \mathcal{D}_k} \tilde{p}_{k,s}$ (Calandriello et al., 2020). Again, due to asynchronous communication, for data point $s$ that was added during the $k'$-th communication, i.e., $s \in \Delta \mathcal{D}_{k'}$, we have $q_{k,s} = q_{k',s}$, $\tilde{p}_{k,s} = \tilde{p}_{k',s}$ and thus $\sum_{s \in \mathcal{D}_k} \tilde{p}_{k,s} = \sum_{k'=1}^k \sum_{s \in \Delta \mathcal{D}_{k'}} \tilde{p}_{k',s}$. Compared with Li et al. (2022); Calandriello et al. (2020) that re-sample all $s \in \mathcal{D}_k$ using $\tilde{p}_{k,s} = \bar{q}\tilde{\sigma}^2_{k-1}(\mathbf{x}_s)$, we use a different approximated variance function for each $\Delta \mathcal{S}_{k'}$. Nevertheless, with our design in Section 4.1, i.e., $\tilde{p}_{k',s} = \bar{q}\tilde{\sigma}^2_{k'-1}(\mathbf{x}_s)$, we show in Lemma 4.3 that we can still ensure $|\mathcal{S}_k| = O(\gamma_T)$, as long as a proper threshold $D$ is chosen to avoid any $\Delta \mathcal{D}_{k'}$ being too large.

**Lemma 4.3** (Bounding $\sum_{k=0}^B \mathbb{P}\left( E_k \cap (\cap_{k=0}^B A_k^C) \right)$). *By setting $\bar{q} = 4 \ln(2\sqrt{2}T/\delta)\beta(1 + \epsilon/3)/\epsilon^2$, and $\lambda \leq k(\mathbf{x}, \mathbf{x})$, $\forall \mathbf{x} \in \mathcal{A}$, we have $\sum_{k=0}^B \mathbb{P}\left( E_k \cap (\cap_{k=0}^B A_k^C) \right) \leq \delta/2$, for $\delta \in (0, 1)$.*

Putting everything together, we have $\mathbb{P}\left( \cup_{k=0}^B H_k \right) \leq \delta$, for $\delta \in (0, 1)$, which finishes the proof. $\quad\square$

## 4.3 Analysis of Regret and Communication Cost

Lemma 4.1 guarantees a compact and accurate dictionary for Nyström approximation throughout the learning process. Based on it, we establish the upper bounds for the cumulative regret and communication cost of Async-KernelUCB. First, motivated by the confidence ellipsoid for asynchronous linear bandits (He et al., 2022), we construct the following confidence ellipsoid for our approximated estimator for kernel bandit defined in Section 3.2 (proof is provided in appendix).

**Lemma 4.4** (Confidence ellipsoid for approximated estimator). *Under the same condition as Lemma 4.1, with probability at least $1 - 2\delta$, for $\delta \in (0, 1)$, we have $\forall k$ that*

$$\|\tilde{\theta}_k - \theta_\star\|_{\tilde{\mathbf{V}}_k} \leq (1/\sqrt{1-\epsilon} + 1)\sqrt{\lambda}S + 2R\left(\sqrt{1 + ND\beta} + N\sqrt{2D\beta}\right)\sqrt{\ln(1/\delta) + \gamma_T} := \alpha,$$

*where $\tilde{\mathbf{V}}_k := \mathbf{P}_{\mathcal{S}_k} \mathbf{\Phi}_{\mathcal{D}_k}^\top \mathbf{\Phi}_{\mathcal{D}_k} \mathbf{P}_{\mathcal{S}_k} + \lambda\mathbf{I}$ and $\gamma_T := \max_{\mathcal{D} \subset \mathcal{A}: |\mathcal{D}| = T} \frac{1}{2} \log \det(\mathbf{K}_{\mathcal{D},\mathcal{D}}/(D\beta\lambda) + \mathbf{I})$ [2] is the maximum information gain after $T$ interactions (Chowdhury & Gopalan, 2017; Li et al., 2022).*

Then based on Lemma 4.4, we establish Theorem 4.5 below (proof is provided in appendix).

**Theorem 4.5.** *Under the same condition as Lemma 4.1, we have*

$$R_T \leq 4N\gamma_T LS + 4\sqrt{2}\Big[(1/\sqrt{1-\epsilon} + 1)\sqrt{\lambda}S + 2R\left(\sqrt{1 + ND\beta} + N\sqrt{2D\beta}\right)\sqrt{\ln(1/\delta) + \gamma_T}\Big]$$
$$\cdot \sqrt{T\beta[1 + N\beta(L^2/\lambda + D)]\gamma_T}$$

*with probability at least $1 - 2\delta$, and*

$$C_T \leq 2\gamma_T(N + 4\beta/D)\big[3(|\mathcal{S}_B|^2 + |\mathcal{S}_B|) + d|\mathcal{S}_B|\big].$$

*where the dictionary size $|\mathcal{S}_B| \leq 12\beta(1 + \beta D)\bar{q}\gamma_T$ due to Lemma 4.1. By setting $D = 1/N^2$, we have $R_T = O\left(N\gamma_T LS + \sqrt{T}(S\sqrt{\gamma_T} + \gamma_T)\right)$, and $C_T = \tilde{O}(N^2\gamma_T^3)$.*

---

[2]As discussed in Li et al. (2022), $\gamma_T$ is problem-dependent, showing how fast kernel's eigenvalues decay. For kernels with exponentially decaying eigenvalues, i.e., $\lambda_m = O(\exp(-m^{\beta_e}))$, for $\beta_e > 0$, $\gamma_T = O(\log^{1+1/\beta_e}(T))$, which includes Gaussian kernel that is widely used for GPs and SVMs. For kernels with polynomially decaying eigenvalues, i.e., $\lambda_m = O(m^{-\beta_p})$, for $\beta_p > 1$, $\gamma_T = O(T^{1/\beta_p} \log^{1-1/\beta_p}(T))$.

## 5 EXPERIMENTS

To validate Async-KernelUCB's effectiveness in reducing communication cost, we performed extensive empirical evaluations on both synthetic and real-world datasets, and reported the results (over 10 runs) in Figure 2. The baselines included in our comparisons are: 1) OneKernelUCB (Valko et al., 2013), it learns a shared kernel bandit model across all clients' aggregated data where data aggregation happens immediately after each new data point becomes available; 2) NKernelUCB, it learns a separate kernel bandit model for each client with no communication; 3) FedGLBUCB (Li & Wang, 2022b), it is a synchronous distributed GLB algorithm; 4) DisLinUCB (Wang et al., 2019), it is a synchronous distributed linear bandit algorithm; 5) FedLinUCB (He et al., 2022), it is an asynchronous distributed linear bandit algorithm; and 6) Approx-DisKernelUCB (Li et al., 2022), it is a synchronous distributed kernel bandit algorithm. For all the kernel bandit algorithms, we used the Gaussian kernel $k(x, y) = \exp(-\gamma \|x - y\|^2)$, where we did a grid search of $\gamma \in \{0.1, 1, 4\}$, and for FedGLBUCB, we used Sigmoid function $\mu(z) = (1 + \exp(-z))^{-1}$ as link function. For all algorithms, instead of using their theoretically derived exploration coefficient $\alpha$, we followed the convention Li et al. (2010a); Zhou et al. (2020) to use grid search for $\alpha$ in $\{0.1, 1, 4\}$.

### 5.1 EXPERIMENT SETUP

**Synthetic dataset** We simulated the distributed bandit setting in Section 3.1, with $d = 20, T = 10^4, N = 10^2$. At each time step $t \in [T]$, client $i_t \in [N]$ selects an arm from candidate set $\mathcal{A}_t$ (with $|\mathcal{A}_t| = 20$), which is uniformly sampled from a $\ell_2$ unit ball. Then the reward is generated using one of the following reward functions: 1) $f_1(\mathbf{x}) = \cos(3\mathbf{x}^\top \theta_\star)$, and 2) $f_2(\mathbf{x}) = (\mathbf{x}^\top \theta_\star)^3 - 3(\mathbf{x}^\top \theta_\star)^2 - (\mathbf{x}^\top \theta_\star) + 3$, where the parameter $\theta_\star$ is uniformly sampled from a $\ell_2$ unit ball and fixed.

**UCI Datasets** We also performed experiments using MagicTelescope and Mushroom from the UCI Machine Learning Repository (Dua & Graff, 2017), which are converted to bandit problem following Filippi et al. (2010). Specifically, we partitioned the dataset into 20 clusters using k-means, and used the centroid of each cluster as the context for the arms and used the averaged response as mean reward (the response is binarized by setting one class as 1, and all the others as 0). Then we simulated the distributed bandit setting in Section 3.1 with $|\mathcal{A}_t| = 20$, $T = 10^4$ and $N = 10^2$.

**MovieLens and Yelp dataset** Yelp dataset is released by the Yelp dataset challenge, and consists of 4.7 million rating entries for 157 thousand restaurants by 1.18 million users. MovieLens consists of 25 million ratings between 160 thousand users and 60 thousand movies (Harper & Konstan, 2015). Following the pre-processing steps in Ban et al. (2021), we built the rating matrix by choosing the top 2,000 users and top 10,000 restaurants/movies and used singular-value decomposition to extract a 10-dimension feature vector for each user and restaurant/movie. We treated ratings greater than 2 as positive, and simulated the distributed bandit setting in Section 3.1 with $T = 10^4$ and $N = 10^2$. The candidate set $\mathcal{A}_t$ (with $|\mathcal{A}_t| = 20$) is constructed by sampling an arm with positive reward and nineteen arms with negative reward from the arm pool, and the concatenation of user and restaurant/movie feature vector is used as the context vector for the arm (thus $d = 20$).

### 5.2 EXPERIMENT RESULTS

OneKernelUCB and NKernelUCB correspond to the two extreme cases where the clients either communicate in every time step to learn a shared model, or they learn their own models independently with no communication. As shown in Figure 2, OneKernelUCB achieved the smallest cumulative regret in almost all experiments, but also incurred the highest communication cost, i.e., $O(TNd)$ due to sending each new data point to all clients in every round, which demonstrates the necessity of communication efficient bandit algorithms. On the other hand, distributed linear bandit algorithms, e.g., DisLinUCB and FedLinUCB, incurred very low communication cost as they directly communicate via the $d \times d$ statistics, but fail to capture the complicated reward mappings in most of these datasets, e.g., in Figure 2(d), they even had much worse regret than NKernelUCB that requries no communication. Equipped with logistic function, distributed GLB algorithm FedGLBUCB attained both low regret and low communication cost on the two classification datasets, i.e., Figure 2(c) and Figure 2(d), but required many iterations of distributed gradient updates to converge on the other four datasets where logistic function may not fit, and led to huge communication costs. In comparison, Approx-DisKernelUCB and our proposed Async-KernelUCB had consistently smaller

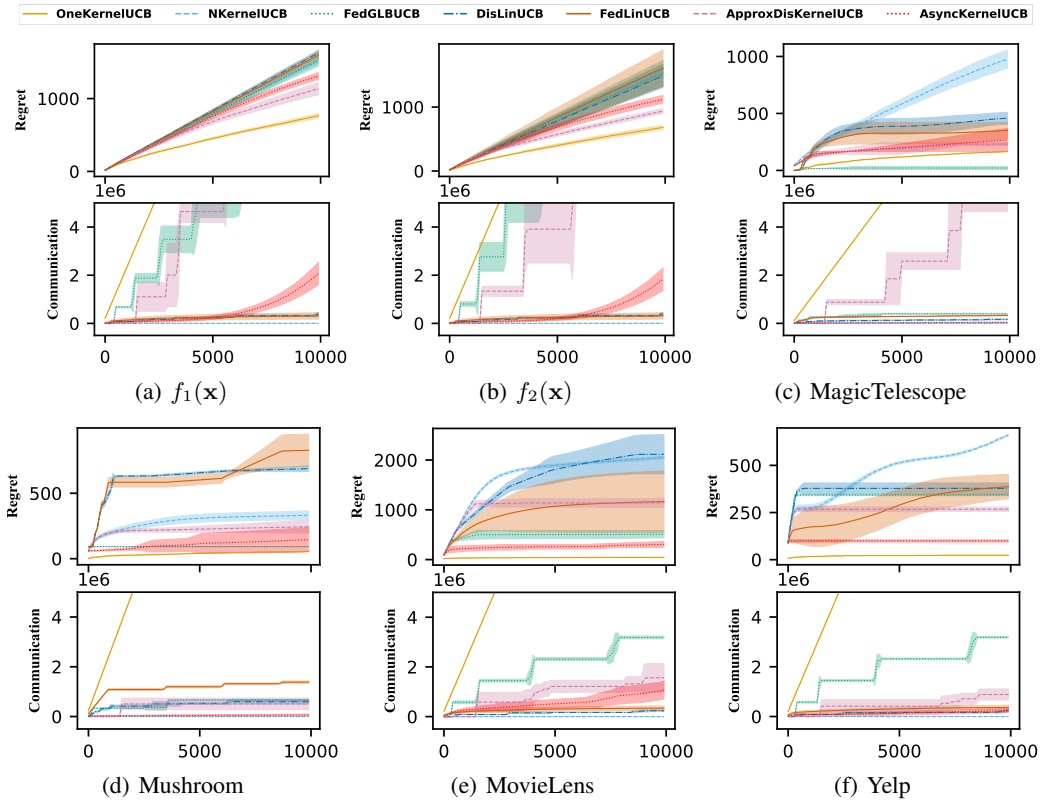

Figure 2: Experiment results on synthetic and real-world datasets.

regret than their linear counterparts, while requiring relatively lower communication cost for joint kernel estimation. It is also worth noting that despite having the same $\tilde{O}(N^2\gamma_T^3)$ theoretical scaling in communication cost, Async-KernelUCB incurs much smaller communication cost empirically, while having comparable or even better regret than Approx-DisKernelUCB.

## CONCLUSIONS

In this paper, we proposed the first asynchronous algorithm for distributed kernel bandit, which relaxes the limitation of prior work that requires impractical global synchronization to update the Nyström embedding function and share the embedded statistics across all clients. To ensure approximation quality and compactness of the constructed dictionary in asynchronous communications, we designed an incremental update procedure tailored to this communication scheme, and a transformation operation on the server side to enable joint kernel estimation using statistics with different embeddings. With the improved robustness against delays and unavailability of clients by having asynchronous communication, we show that to attain near-optimal regret, the proposed algorithm still only incurs an $\tilde{O}(N^2\gamma_T^3)$ communication cost, matching that of the prior work.

The lower bound analysis for the communication cost of distributed contextual bandits still remains an open problem, and is an important future direction. To the best of our knowledge, the only applicable lower bound states that, in order to have smaller regret than the trivial $O(\sqrt{NT})$ result, i.e., run $N$ instances of optimal bandit algorithm with no communication, $\Omega(N)$ communications is necessary (He et al., 2022). In comparison, it is more interesting to know what the communication lower bound is in order to attain the optimal $O(\sqrt{T})$ regret. Moreover, motivated by the differential private (DP) version of DisLinUCB by Dubey & Pentland (2020), i.e., apply randomized mechanisms to the shared sufficient statistics, another interesting direction is a DP version of our Async-KernelUCB, in which case, the main focus is a privacy-preserving construction of the shared embedding function.

## ACKNOWLEDGEMENT

Hongning Wang acknowledges the support by NSF grants IIS-2213700, IIS-2128019 and IIS-1838615. Mengdi Wang acknowledges the support by NSF grants DMS-1953686, IIS-2107304, CMMI1653435, ONR grant 1006977, and http://C3.AI.

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

# A    Technical Lemmas

**Lemma A.1** (Lemma 11 of Abbasi-Yadkori et al. (2011))**.** *Let $\{\mathbf{x}_t\}_{t=1}^{\infty}$ be a sequence in $\mathbb{R}^d$, $V \in \mathbb{R}^{d \times d}$ a positive definite matrix, and define $V_t = V + \sum_{s=1}^{t} \mathbf{x}_s \mathbf{x}_s^{\top}$. Then we have that*

$$\ln\big(\frac{\det(V_n)}{\det(V)}\big) \leq \sum_{t=1}^{n} \|\mathbf{x}_t\|_{V_{t-1}^{-1}}^2.$$

*If $\|\mathbf{x}_t\|_2 \leq L, \forall t$, and $\lambda_{\min}(V) \geq \max(1, L^2)$, then*

$$\sum_{t=1}^{n} \|\mathbf{x}_t\|_{V_{t-1}^{-1}}^2 \leq 2\ln\big(\frac{\det(V_n)}{\det(V)}\big).$$

**Lemma A.2** (Lemma 12 of Abbasi-Yadkori et al. (2011))**.** *Let $A$, $B$ and $C$ be positive semi-definite matrices such that $A = B + C$. Then, we have that:*

$$\sup_{\mathbf{x} \neq \mathbf{0}} \frac{\mathbf{x}^{\top} A \mathbf{x}}{\mathbf{x}^{\top} B \mathbf{x}} \leq \frac{\det(A)}{\det(B)}$$

**Lemma A.3** (Lemma A.2 of Li et al. (2022))**.** *Define positive definite matrices $A = \lambda \mathbf{I} + \boldsymbol{\Phi}_1^{\top} \boldsymbol{\Phi}_1 + \boldsymbol{\Phi}_2^{\top} \boldsymbol{\Phi}_2$ and $B = \lambda \mathbf{I} + \boldsymbol{\Phi}_1^{\top} \boldsymbol{\Phi}_1$, where $\boldsymbol{\Phi}_1^{\top} \boldsymbol{\Phi}_1, \boldsymbol{\Phi}_2^{\top} \boldsymbol{\Phi}_2 \in \mathbb{R}^{p \times p}$ and $p$ is possibly infinite. Then, we have that:*

$$\sup_{\phi \neq \mathbf{0}} \frac{\phi^{\top} A \phi}{\phi^{\top} B \phi} \leq \frac{\det(\mathbf{I} + \lambda^{-1} \mathbf{K}_A)}{\det(\mathbf{I} + \lambda^{-1} \mathbf{K}_B)}$$

*where $\mathbf{K}_A = \begin{bmatrix} \boldsymbol{\Phi}_1 \\ \boldsymbol{\Phi}_2 \end{bmatrix} \begin{bmatrix} \boldsymbol{\Phi}_1^{\top}, \boldsymbol{\Phi}_2^{\top} \end{bmatrix}$ and $\mathbf{K}_B = \boldsymbol{\Phi}_1 \boldsymbol{\Phi}_1^{\top}$.*

**Lemma A.4** (Eq (26) and Eq (27) of Zenati et al. (2022))**.** *Let $\{\phi_t\}_{t=1}^{\infty}$ be a sequence in $\mathbb{R}^p$, $V \in \mathbb{R}^{p \times p}$ a positive definite matrix, where $p$ is possibly infinite, and define $V_t = V + \sum_{s=1}^{t} \phi_s \phi_s^{\top}$. Then we have that*

$$\sum_{t=1}^{n} \min\big(\|\phi_t\|_{V_{t-1}^{-1}}^2, 1\big) \leq 2\ln\big(\det(\mathbf{I} + \lambda^{-1} \mathbf{K}_{V_t})\big),$$

*where $\mathbf{K}_{V_t}$ is the kernel matrix corresponding to $V_t$ as defined in Lemma A.3.*

**Lemma A.5** (Lemma 4 of Calandriello et al. (2020))**.** *For $t > t'$, we have for any $\mathbf{x} \in \mathbb{R}^d$*

$$\sigma_t^2(\mathbf{x}) \leq \sigma_{t'}^2(\mathbf{x}) \leq \big(1 + \sum_{s=t'+1}^{t} \sigma_{t'}^2(\mathbf{x}_s)\big) \sigma_t^2(\mathbf{x})$$

**Lemma A.6** (Lemma 6 of Calandriello et al. (2019))**.** *If $\mathcal{S}_k$ is $\epsilon$-accurate w.r.t. $\mathcal{D}_k$, then*

$$\frac{1 - \epsilon}{1 + \epsilon} \sigma^2(\mathbf{x}) \leq \min(\tilde{\sigma}_k^2(\mathbf{x}), 1) \leq \frac{1 + \epsilon}{1 - \epsilon} \sigma^2(\mathbf{x})$$

*for all $\mathbf{x} \in \mathbb{R}^d$.*

**Lemma A.7** (Proposition 7 of Calandriello et al. (2019))**.** *Let $G_1, \ldots, G_n$ be a sequence of independent self-adjoint random operators such that $\mathbb{E}[G_i] = 0$ and $\|G_i\| \leq R$. Then for any $\epsilon \geq 0$, we have*

$$\mathbb{P}\big(\|\sum_{i=1}^{t} G_i\| \geq \epsilon\big) \leq 4t \exp\big(-\frac{\epsilon^2/2}{\|\sum_{i=1}^{t} \mathbb{E}[G_i^2]\| + R\epsilon/3}\big).$$

**Lemma A.8** (Proposition 8 of (Calandriello et al., 2019))**.** *Let $\{q_s\}_{s=1}^{t}$ be independent Bernoulli random variables, each with success probability $p_s$. Then we have*

$$\mathbb{P}\big(\sum_{s=1}^{t} q_s \geq 3 \sum_{s=1}^{t} p_s\big) \leq \exp(-2 \sum_{s=1}^{t} p_s).$$

**Lemma A.9** (Corollary 7.7.4. (a) of Horn & Johnson (2012))**.** *Let $A, B$ be positive definite matrices, such that $A \succeq B$, then we have*

$$A^{-1} \preceq B^{-1}.$$

**Lemma A.10** (Lemma 2.2 of Tie et al. (2011))**.** *For any positive semi-definite matrices $A, B$ and $C$, it holds that $\det(A + B + C) + \det(A) \geq \det(A + B) + \det(A + C)$.*

# B    PROOF OF LEMMAS IN SECTION 4.2

Let's define the unfavorable event $H_k = A_k \cup E_k$, where $A_k$ is the event that the dictionary $\mathcal{S}_k$ is not $\epsilon$-accurate w.r.t. $\mathcal{D}_k$, and $E_k$ is the event that the size of dictionary $|\mathcal{S}_k|$ is large, i.e., $|\mathcal{S}_k| > 12\beta(1 + \beta D)\bar{q}\gamma_T$. Therefore, we want to bound the probability of $\cup_{k=0}^B H_k$, which can be decomposed as

$$\mathbb{P}\big(\cup_{k=0}^B H_k\big) = \mathbb{P}\big(\cup_{k=0}^B (A_k \cup E_k)\big) = \mathbb{P}\big((\cup_{k=0}^B A_k) \cup (\cup_{k=0}^B E_k)\big)$$
$$= \mathbb{P}\big(\cup_{k=0}^B A_k\big) + \mathbb{P}\big(\cup_{k=0}^B E_k\big) - \mathbb{P}\big((\cup_{k=0}^B A_k) \cap (\cup_{k=0}^B E_k)\big)$$
$$= \mathbb{P}\big(\cup_{k=0}^B A_k\big) + \mathbb{P}\big((\cup_{k=0}^B E_k) \cap (\cup_{k=0}^B A_k)^C\big)$$

Note that, as in Calandriello et al. (2017), we bound the second term as $\mathbb{P}\big((\cup_{k=0}^B E_k) \cap (\cup_{k=0}^B A_k)^C\big) = \mathbb{P}\big((\cup_{k=0}^B E_k) \cap (\cap_{k=0}^B A_k^C)\big) = \mathbb{P}\big(\cup_{k=0}^B [E_k \cap (\cap_{k=0}^B A_k^C)]\big) \leq \sum_{k=0}^B \mathbb{P}\big(E_k \cap (\cap_{k=0}^B A_k^C)\big)$. For the first term $\mathbb{P}\big(\cup_{k=0}^B A_k\big)$, we need a decomposition different from prior works (Calandriello et al., 2017; 2019), since our dictionary is *incrementally updated with a batch of samples at each communication round* (line 9 in Algorithm 1). Specifically, when bounding the probability of having an inaccurate dictionary at the $k$-th communication, i.e., event $A_k$, we need to condition on the event that dictionaries at all previous communications are $\epsilon$-accurate, i.e., event $\cap_{k'=0}^{k-1} A_{k'}^C$. Hence, we decompose $\mathbb{P}\big(\cup_{k=0}^B A_k\big) = 1 - \mathbb{P}\big(\cap_{k=0}^B A_k^C\big) = 1 - \mathbb{P}(A_0^C) \prod_{k=1}^B \mathbb{P}\big(A_k^C | \cap_{k'=0}^{k-1} A_{k'}^C\big) = 1 - \prod_{k=1}^B [1 - \mathbb{P}\big(A_k | \cap_{k'=0}^{k-1} A_{k'}^C\big)] \leq \sum_{k=1}^B \mathbb{P}\big(A_k | \cap_{k'=0}^{k-1} A_{k'}^C\big)$, where the second equality is due to Bayes theorem, the third equality is because $\mathcal{D}_0 = \emptyset$ is well-approximated by $\mathcal{S}_0 = \emptyset$, and thus $\mathbb{P}\big(A_0^C\big) = 1$, and the inequality is due to Weierstrass product inequality. Putting everything together, we have

$$\mathbb{P}\big(\cup_{k=0}^B H_k\big) \leq \sum_{k=1}^B \mathbb{P}\big(A_k | \cap_{k'}^{k-1} A_{k'}^C\big) + \sum_{k=1}^B \mathbb{P}\big(E_k \cap (\cap_{k=0}^B A_k^C)\big) \tag{7}$$

Then we can upper bound these two terms using Lemma 4.2 and Lemma 4.3 given in Section 4.2, which leads to $\mathbb{P}\big(\cup_{k=0}^B H_k\big) \leq \delta$, for $\delta \in (0,1)$, and thus finishes the proof of Lemma 4.1.

*Proof of Lemma 4.2: bounding* $\sum_{k=1}^B \mathbb{P}\big(A_k | \cap_{k'}^{k-1} A_{k'}^C\big)$. As Calandriello et al. (2019), we can rewrite the event $A_k$, based on the definition of $\epsilon$-accuracy given in equation 6, as

$$A_k = \big\{ \|\sum_{s \in \mathcal{D}_k} G_{k,s}\| > \epsilon \big\}$$

where $G_{k,s} = (\frac{q_{k,s}}{\tilde{p}_{k,s}} - 1)\psi_{k,s}\psi_{k,s}^\top$ and $\psi_{k,s} = (\mathbf{\Phi}_{\mathcal{D}_k}^\top \mathbf{\Phi}_{\mathcal{D}_k} + \lambda\mathbf{I})^{-1/2}\phi(\mathbf{x}_s)$. Then let's define $\mathcal{F}_k := \{q_{k,s}, \eta_s\}_{s \in \mathcal{D}_k}$ for $k \in [B]$, which contains all randomness in the construction of $\mathcal{S}_k$ during the $k$-th communication. With conditioning, we have

$$\mathbb{P}(A_k | \cap_{k'}^{k-1} A_{k'}^C) = \mathbb{P}\big(\|\sum_{s \in \mathcal{D}_k} G_{k,s}\| > \epsilon \,|\, \cap_{k'}^{k-1} A_{k'}^C\big) = \mathbb{E}_{\mathcal{F}_k}\big[\mathbf{1}\{\|\sum_{s \in \mathcal{D}_k} G_{k,s}\| > \epsilon\} \,|\, \cap_{k'}^{k-1} A_{k'}^C\big]$$

$$= \mathbb{E}_{\mathcal{F}_{k-1}}\big[\mathbb{E}_{\mathcal{F}_k \backslash \mathcal{F}_{k-1}}\big[\mathbf{1}\{\|\sum_{s \in \mathcal{D}_k} G_{k,s}\| > \epsilon\} | \mathcal{F}_{k-1}\big] \,|\, \cap_{k'}^{k-1} A_{k'}^C\big]$$

$$= \mathbb{E}_{\mathcal{F}_{k-1}:\cap_{k'}^{k-1} A_{k'}^C}\big[\mathbb{E}_{\mathcal{F}_k \backslash \mathcal{F}_{k-1}}\big[\mathbf{1}\{\|\sum_{s \in \mathcal{D}_k} G_{k,s}\| > \epsilon\} | \mathcal{F}_{k-1}\big]\big]$$

$$= \mathbb{E}_{\mathcal{F}_{k-1}:\cap_{k'}^{k-1} A_{k'}^C}\big[\mathbb{P}_{\mathcal{F}_k \backslash \mathcal{F}_{k-1}}\big(\|\sum_{s \in \mathcal{D}_k} (\frac{q_{k,s}}{\tilde{p}_{k,s}} - 1)\psi_{k,s}\psi_{k,s}^\top\| > \epsilon \,|\, \mathcal{F}_{k-1}\big)\big].$$

where the third equality holds because when conditioned on the event $\cap_{k'}^{k-1} A_{k'}^C$, the outcomes associated with the complement of this event have zero probability, and thus we can restrict the expectation to the outcomes where the event $\cap_{k'}^{k-1} A_{k'}^C$ holds.

Consider the $k$-th communication for $k \in [B]$. We denote the client who triggers the $k$-th communication as $c_k \in [N]$, and the time step when the $k$-th communication happens as $t_k \in [T]$. In addition, recall that we denote the sequence of time steps in-between client $c_k$'s last communication (whose index is denoted as $\underline{k}(c_k) \in [0, k-1]$) and the current (the $k$-th) communication when client $c_k$'s is active as $\Delta\mathcal{D}_k := \mathcal{N}_{t_k}(c_k) \backslash \mathcal{N}_{t_{\underline{k}(c_k)}}(c_k) = \{t_{\underline{k}(c_k)} < s \leq t_k : i_s = c_k\}$.

Note that due to our incremental update procedure, for some data point with time index $s$, that was added into $\mathcal{D}_k$ during the $k'$-th communication (sent to the server in the form of embedded statistics), i.e., $s \in \Delta\mathcal{D}_{k'}$, for $k' = 1, \ldots, k$, we have $q_{k,s} = q_{k',s}$ and $\tilde{p}_{k,s} = \tilde{p}_{k',s}$. When conditioned on $\mathcal{F}_{k-1}$, $q_{k,s}$ for all $s \in \mathcal{D}_k$ are independent Bernoulli random variable with mean $\tilde{p}_{k,s}$, because they only correlate via the approximated variance function(s) that were used for arm selection and RLS sampling up to the $k$-th communication, which are deterministic conditioned on $\mathcal{F}_{k-1}$, and thus both $\tilde{p}_{k,s}$ and $\psi_{k,s}$ are deterministic as well.

Therefore, we can bound $\mathbb{P}_{\mathcal{F}_k \backslash \mathcal{F}_{k-1}}\big(\|\sum_{s \in \mathcal{D}_k}(\frac{q_{k,s}}{\tilde{p}_{k,s}} - 1)\psi_{k,s}\psi_{k,s}^\top\| > \epsilon | \mathcal{F}_{k-1}\big)$ using Lemma A.7. First, we need to show that each term in the summation has zero mean and bounded norm, i.e., $\mathbb{E}_{\mathcal{F}_k \backslash \mathcal{F}_{k-1}}[G_{k,s} | \mathcal{F}_{k-1}] = 0$ and $\|G_{k,s}\| \leq R$ for some constant $R$:

$$\mathbb{E}_{\mathcal{F}_k \backslash \mathcal{F}_{k-1}}\big[(\frac{q_{k,s}}{\tilde{p}_{k,s}} - 1)\psi_{k,s}\psi_{k,s}^\top | \mathcal{F}_{k-1}\big] = (\frac{\mathbb{E}_{\mathcal{F}_k \backslash \mathcal{F}_{k-1}}[q_{k,s} | \mathcal{F}_{k-1}]}{\tilde{p}_{k,s}} - 1)\psi_{k,s}\psi_{k,s}^\top = 0,$$

and

$$\|G_{k,s}\| = \|(\frac{q_{k,s}}{\tilde{p}_{k,s}} - 1)\psi_{k,s}\psi_{k,s}^\top\| \leq (\frac{q_{k,s}}{\tilde{p}_{k,s}} - 1)\|\psi_{k,s}\psi_{k,s}^\top\| \leq \frac{\sigma_k^2(\mathbf{x}_s)}{\tilde{p}_{k,s}},$$

where the last inequality is because $q_{k,s} \leq 1$ and $\|\psi_{k,s}\psi_{k,s}^\top\| = \psi_{k,s}^\top\psi_{k,s} = \sigma_k^2(\mathbf{x}_s)$. As mentioned earlier, for $s \in \Delta\mathcal{D}_{k'}$, $k' = 1, \ldots, k$, we have $\tilde{p}_{k,s} = \tilde{p}_{k',s} = \bar{q}\tilde{\sigma}_{k'-1}^2(\mathbf{x}_s)$, i.e., during the $k'$-th communication, client $c_{k'}$ first receives server's latest statistics to compute $\tilde{\sigma}_{k'-1}^2(\cdot)$ for RLS sampling. Conditioned on $\cap_{k'=0}^k A_{k'}^C$ and by Lemma A.6, we have $\tilde{\sigma}_{k'-1}^2(\mathbf{x}_s) \geq \sigma_{k'-1}^2(\mathbf{x}_s)/\beta$, where $\beta := (1 + \epsilon)/(1 - \epsilon)$. Hence,

$$\|G_{k,s}\| \leq \frac{\sigma_k^2(\mathbf{x}_s)}{\tilde{p}_{k,s}} = \frac{\sigma_k^2(\mathbf{x}_s)}{\bar{q}\tilde{\sigma}_{k'-1}^2(\mathbf{x}_s)} \leq \frac{\beta}{\bar{q}}\frac{\sigma_k^2(\mathbf{x}_s)}{\sigma_{k'-1}^2(\mathbf{x}_s)} \leq \frac{\beta}{\bar{q}} := R.$$

where the last inequality is because the variance is non-increasing over time. Then by Lemma A.7,

$$\mathbb{P}_{\mathcal{F}_k \backslash \mathcal{F}_{k-1}}\big(\|\sum_{s \in \mathcal{D}_k} G_{k,s}\| > \epsilon | \mathcal{F}_{k-1}\big) \leq 4|\mathcal{D}_k| \exp\big(-\frac{\epsilon^2/2}{\|\sum_{s \in \mathcal{D}_k} \mathbb{E}_{\mathcal{F}_k \backslash \mathcal{F}_{k-1}}[G_{k,s}^2 | \mathcal{F}_{k-1}]\| + R\epsilon/3}\big)$$

Now we need to further upper bound the term $\|\sum_{s \in \mathcal{D}_k} \mathbb{E}_{\mathcal{F}_k \backslash \mathcal{F}_{k-1}}[G_{k,s}^2 | \mathcal{F}_{k-1}]\|$. First, note that

$$\mathbb{E}_{\mathcal{F}_k \backslash \mathcal{F}_{k-1}}[G_{k,s}^2 | \mathcal{F}_{k-1}] = \mathbb{E}_{\mathcal{F}_k \backslash \mathcal{F}_{k-1}}\big[(\frac{q_{k,s}}{\tilde{p}_{k,s}} - 1)^2 \psi_{k,s}\psi_{k,s}^\top\psi_{k,s}\psi_{k,s}^\top | \mathcal{F}_{k-1}\big]$$

$$= \mathbb{E}_{\mathcal{F}_k \backslash \mathcal{F}_{k-1}}\big[(\frac{q_{k,s}}{\tilde{p}_{k,s}} - 1)^2 | \mathcal{F}_{k-1}\big]\psi_{k,s}\psi_{k,s}^\top\psi_{k,s}\psi_{k,s}^\top,$$

and $\mathbb{E}_{\mathcal{F}_k \backslash \mathcal{F}_{k-1}}[(\frac{q_{k,s}}{\tilde{p}_{k,s}} - 1)^2 | \mathcal{F}_{k-1}] = \mathbb{E}_{\mathcal{F}_k \backslash \mathcal{F}_{k-1}}[(\frac{q_{k,s}}{\tilde{p}_{k,s}})^2 | \mathcal{F}_{k-1}] - 2\mathbb{E}_{\mathcal{F}_k \backslash \mathcal{F}_{k-1}}[\frac{q_{k,s}}{\tilde{p}_{k,s}} | \mathcal{F}_{k-1}] + 1 = \mathbb{E}_{\mathcal{F}_k \backslash \mathcal{F}_{k-1}}[\frac{q_{k,s}}{\tilde{p}_{k,s}^2} | \mathcal{F}_{k-1}] - 1 = \frac{1}{\tilde{p}_{k,s}} - 1 \leq \frac{1}{\tilde{p}_{k,s}}$. Substituting this to the RHS, we have

$$\mathbb{E}_{\mathcal{F}_k \backslash \mathcal{F}_{k-1}}[G_{k,s}^2 | \mathcal{F}_{k-1}] \preceq \frac{1}{\tilde{p}_{k,s}}\psi_{k,s}\psi_{k,s}^\top\psi_{k,s}\psi_{k,s}^\top \preceq \frac{1}{\tilde{p}_{k,s}}\|\psi_{k,s}\psi_{k,s}^\top\|\psi_{k,s}\psi_{k,s}^\top \preceq R\psi_{k,s}\psi_{k,s}^\top,$$

and thus,

$$\|\sum_{s \in \mathcal{D}_k} \mathbb{E}_{\mathcal{F}_k \backslash \mathcal{F}_{k-1}}[G_{k,s}^2 | \mathcal{F}_{k-1}]\| \leq R\|\sum_{s \in \mathcal{D}_k} \psi_{k,s}\psi_{k,s}^\top\|$$

$$= R\|\sum_{s \in \mathcal{D}_k}(\mathbf{\Phi}_{\mathcal{D}_k}^\top\mathbf{\Phi}_{\mathcal{D}_k} + \lambda\mathbf{I})^{-1/2}\phi_s\phi_s^\top(\mathbf{\Phi}_{\mathcal{D}_k}^\top\mathbf{\Phi}_{\mathcal{D}_k} + \lambda\mathbf{I})^{-1/2}\|$$

$$= R\|(\mathbf{\Phi}_{\mathcal{D}_k}^\top\mathbf{\Phi}_{\mathcal{D}_k} + \lambda\mathbf{I})^{-1/2}\mathbf{\Phi}_{\mathcal{D}_k}^\top\mathbf{\Phi}_{\mathcal{D}_k}(\mathbf{\Phi}_{\mathcal{D}_k}^\top\mathbf{\Phi}_{\mathcal{D}_k} + \lambda\mathbf{I})^{-1/2}\| \leq R,$$

where the first equality is by definition of $\psi_{k,s}$. Putting everything together, we have

$$\mathbb{P}_{\mathcal{F}_k \backslash \mathcal{F}_{k-1}}\big(\|\sum_{s \in \mathcal{D}_k} G_{k,s}\| > \epsilon | \mathcal{F}_{k-1}\big) \leq 4|\mathcal{D}_k| \exp\big(-\frac{\epsilon^2/2}{1 + \epsilon/3} \cdot \frac{\bar{q}}{\beta}\big),$$

and thus $\mathbb{P}(A_k \mid \cap_{k'=0}^{k-1} A_{k'}^C) \leq 4|\mathcal{D}_k| \exp(-\frac{\epsilon^2/2}{1+\epsilon/3} \cdot \frac{\bar{q}}{\beta})$. Summing over $B$ terms, we have

$$\sum_{k=0}^{B} \mathbb{P}(A_k| \cap_{k'=0}^{k-1} A_{k'}^C) \leq 4\exp(-\frac{\epsilon^2/2}{1+\epsilon/3} \cdot \frac{\bar{q}}{\beta}) \sum_{k=1}^{B} |\mathcal{D}_k| \leq 4T^2 \exp(-\frac{\epsilon^2/2}{1+\epsilon/3} \cdot \frac{\bar{q}}{\beta})$$

In order to make sure $\sum_{k=0}^{B} \mathbb{P}(A_k| \cap_{k'=0}^{k-1} A_{k'}^C) \leq \frac{\delta}{2}$, we need to set $\bar{q} = 4\beta \frac{1+\epsilon/3}{\epsilon^2} \ln(\frac{2\sqrt{2}T}{\delta})$.

$\square$

*Proof of Lemma 4.3: bounding $\sum_{k=0}^{B} \mathbb{P}(E_k \cap (\cap_{k=0}^{B} A_k^C))$.* First, note that $\mathbb{P}(E_0 \cap (\cap_{k=0}^{B} A_k^C)) = 0$, because $\mathcal{S}_0 = \emptyset$, and by definition of $q_{k,s}$ for $s \in \mathcal{D}_k$, the size of dictionary $|\mathcal{S}_k| = \sum_{s \in \mathcal{D}_k} q_{k,s}$. We formally define unfavorable event $E_k$ as

$$E_k = \{ \sum_{s \in \mathcal{D}_k} q_{k,s} > 12\beta(1+\beta D)\bar{q}\gamma_T \},$$

where $\beta = (1+\epsilon)/(1-\epsilon)$. Similar to Calandriello et al. (2017; 2019), we will use a stochastic dominance argument to upper bound the probability of event $E_k$. First, we use conditioning again to rewrite $\mathbb{P}(E_k \cap (\cap_{k=1}^{B} A_k^C))$ as

$$\mathbb{P}(E_k \cap (\cap_{k=1}^{B} A_k^C)) = \mathbb{P}(E_k \mid \cap_{k=1}^{B} A_k^C)\mathbb{P}(\cap_{k=1}^{B} A_k^C) \leq \mathbb{P}(E_k \mid \cap_{k=1}^{B} A_k^C)$$

$$= \mathbb{P}( \sum_{s \in \mathcal{D}_k} q_{k,s} \geq 12\beta(1+\beta D)\bar{q}\gamma_T \mid \cap_{k=1}^{B} A_k^C)$$

$$= \mathbb{E}_{\mathcal{F}_{k-1}:\cap_{k=1}^{B} A_k^C} \left[ \mathbb{P}_{\mathcal{F}_k \backslash \mathcal{F}_{k-1}}( \sum_{s \in \mathcal{D}_k} q_{k,s} \geq 12\beta(1+\beta D)\bar{q}\gamma_T \mid \mathcal{F}_{k-1}) \right].$$

As discussed earlier, when conditioned on $\mathcal{F}_{k-1}$, $q_{k,s}$ for $s \in \mathcal{D}_k$ becomes independent Bernoulli random variable, with mean $\tilde{p}_{k,s}$. In addition, as a result of our incremental dictionary update (line 9 in Algorithm 1), the partition in $\mathcal{D}_k$ that were added during the $k'$-th communication for $k' \in 1, \ldots, k$, which is denoted by $\Delta\mathcal{D}_{k'}$, is sampled using $\bar{q}\tilde{\sigma}_{k'-1}^2(\mathbf{x}_s)$ for $s \in \Delta\mathcal{D}_{k'}$. Hence,

$$\mathbb{E}_{\mathcal{F}_k \backslash \mathcal{F}_{k-1}} \left[ \sum_{s \in \mathcal{D}_k} q_{k,s}|\mathcal{F}_{k-1} \right] = \sum_{s \in \mathcal{D}_k} \tilde{p}_{k,s}$$

$$= \sum_{k'=1}^{k} \sum_{s \in \Delta\mathcal{D}_{k'}} \tilde{p}_{k',s} = \bar{q} \sum_{k'=1}^{k} \sum_{s \in \Delta\mathcal{D}_{k'}} \tilde{\sigma}_{k'-1}^2(\mathbf{x}_s)$$

$$\leq \beta\bar{q} \sum_{k'=1}^{k} \sum_{s \in \Delta\mathcal{D}_{k'}} \sigma_{k'-1}^2(\mathbf{x}_s) = \beta\bar{q} \sum_{k'=1}^{k} \sum_{s \in \Delta\mathcal{D}_{k'}} \sigma_{k'-1,s-1}^2(\mathbf{x}_s) \cdot \frac{\sigma_{k'-1}^2(\mathbf{x}_s)}{\sigma_{k'-1,s-1}^2(\mathbf{x}_s)}$$

$$\leq \beta\bar{q} \sum_{k'=1}^{k} \sum_{s \in \Delta\mathcal{D}_{k'}} \sigma_{k'-1,s-1}^2(\mathbf{x}_s) \cdot [1 + \sum_{s' \in \Delta\mathcal{D}_{k'}:s' \leq s-1} \sigma_{k'-1}^2(\mathbf{x}_{s'})]$$

$$\leq \beta\bar{q} \sum_{k'=1}^{k} \sum_{s \in \Delta\mathcal{D}_{k'}} \sigma_{k'-1,s-1}^2(\mathbf{x}_s) \cdot [1 + \sum_{s' \in \Delta\mathcal{D}_{k'}:s' \leq s-1} \sigma_{\underline{k}'(c_{k'})}^2(\mathbf{x}_{s'})]$$

$$\leq \beta\bar{q} \sum_{k'=1}^{k} \sum_{s \in \Delta\mathcal{D}_{k'}} \sigma_{k'-1,s-1}^2(\mathbf{x}_s) \cdot [1 + \beta \sum_{s' \in \Delta\mathcal{D}_{k'}:s' \leq s-1} \tilde{\sigma}_{\underline{k}'(c_{k'})}^2(\mathbf{x}_{s'})]$$

$$\leq \beta(1+\beta D)\bar{q} \sum_{k'=1}^{k} \sum_{s \in \Delta\mathcal{D}_{k'}} \sigma_{k'-1,s-1}^2(\mathbf{x}_s)$$

where the imaginary variance function $\sigma_{k'-1,s-1}^2(\cdot)$ is constructed using dataset $(\cup_{k=1}^{k'-1}\Delta\mathcal{D}_k) \cup \{s' \in \Delta\mathcal{D}_{k'} : s' \leq s-1\}$ (not computed in the actual algorithm); the first and forth inequality is due to Lemma A.6 as we conditioned on $\cap_{k=0}^{B} A_k^C$; the second is due to Lemma A.5; the third is because $\underline{k}'(c_{k'}) \leq k'-1$ and the variance is non-increasing over time; and the fifth is due to our event-trigger design in equation 4, i.e., $\sum_{s \in \Delta\mathcal{D}_{k'}:s \leq t_{k'}-1} \tilde{\sigma}_{\underline{k}'(c_{k'})}^2(\mathbf{x}_s) < D$.

Now for each term in the summation on the RHS of the inequality above, we introduce an independent Bernoulli random variable $\hat{q}_{k,s} \sim \mathcal{B}\big(\beta(1+\beta D)\bar{q}\sigma^2_{k'-1,s-1}(\mathbf{x}_s)\big)$. Since $\hat{q}_{k,s}$ stochastically dominates $q_{k,s}$, i.e., $\mathbb{E}\big[q_{k,s} \mid \mathcal{F}_{k-1}\big] = \tilde{p}_{k,s} \leq \beta(1+\beta D)\bar{q}\sigma^2_{k'-1,s-1}(\mathbf{x}_s) = \mathbb{E}\big[\hat{q}_{k,s}\big]$, we have

$$\mathbb{P}\big(\sum_{s\in\mathcal{D}_k} q_{k,s} > 12\beta(1+\beta D)\bar{q}\gamma_T \mid \mathcal{F}_{k-1}\big) \leq \mathbb{P}\big(\sum_{s\in\mathcal{D}_k} \hat{q}_{k,s} > 12\beta(1+\beta D)\bar{q}\gamma_T\big).$$

Then we can further upper bound the RHS

$$\mathbb{P}\big(\sum_{s\in\mathcal{D}_k} \hat{q}_{k,s} > 12\beta(1+\beta D)\bar{q}\gamma_T\big)$$

$$\leq \mathbb{P}\big(\sum_{s\in\mathcal{D}_k} \hat{q}_{k,s} > 3\beta(1+\beta D)\bar{q}\sum_{k'=1}^{k}\sum_{s\in\Delta\mathcal{D}_{k'}} \sigma^2_{k'-1,s-1}(\mathbf{x}_s)\big)$$

$$\leq \exp\big(-2\beta(1+\beta D)\bar{q}\sum_{k'=1}^{k}\sum_{s\in\Delta\mathcal{D}_{k'}} \sigma^2_{k'-1,s-1}(\mathbf{x}_s)\big)$$

where the first inequality is because $\sum_{k'=1}^{k}\sum_{s\in\Delta\mathcal{D}_{k'}} \sigma^2_{k'-1,s-1}(\mathbf{x}_s) \leq 4\gamma_T$, and the second inequality is due to Lemma A.8. By substituting $\bar{q} = 4\beta\frac{1+\epsilon/3}{\epsilon^2}\ln(\frac{2\sqrt{2}T}{\delta})$ and under the condition that $\sum_{k'=1}^{k}\sum_{s\in\Delta\mathcal{D}_{k'}} \sigma^2_{k'-1,s-1}(\mathbf{x}_s) \geq 1$, we have $\exp\big(-2\beta(1+\beta D)\bar{q}\sum_{k'=1}^{k}\sum_{s\in\Delta\mathcal{D}_{k'}} \sigma^2_{k'-1,s-1}(\mathbf{x}_s)\big) \leq \exp\big(-\ln(8T^2/\delta)\big)$. To ensure $\sum_{k'=1}^{k}\sum_{s\in\Delta\mathcal{D}_{k'}} \sigma^2_{k'-1,s-1}(\mathbf{x}_s) \geq 1$, we can set $\lambda \leq k(\mathbf{x},\mathbf{x}), \forall\mathbf{x} \in \mathcal{A}$. Finally, by summing over $B$ terms, we have

$$\sum_{k=0}^{B} \mathbb{P}\big(E_k \cap (\cap_{k=0}^{B} A_k^C)\big) \leq T\exp\big(-\ln(8T^2/\delta)\big) \leq T\cdot\frac{\delta}{8T^2} < \frac{\delta}{2}$$

where the last inequality is because $T \geq 1$.

$\square$

## C  PROOF OF LEMMA 4.4 IN SECTION 4.3

Recall from Section 3.2 that the approximated kernel Ridge regression estimator for $\theta_\star$ is defined as

$$\tilde{\theta}_k = \tilde{\mathbf{V}}_k^{-1}\mathbf{P}_{\mathcal{S}_k}\mathbf{\Phi}_{\mathcal{D}_k}^{\top}\mathbf{y}_{\mathcal{D}_k}$$

where $\tilde{\mathbf{V}}_k := \mathbf{P}_{\mathcal{S}_k}\mathbf{\Phi}_{\mathcal{D}_k}^{\top}\mathbf{\Phi}_{\mathcal{D}_k}\mathbf{P}_{\mathcal{S}_k} + \lambda\mathbf{I}$. Then we can decompose

$$\|\tilde{\theta}_k - \theta_\star\|^2_{\tilde{\mathbf{V}}_k} = (\tilde{\theta}_k - \theta_\star)^{\top}\tilde{\mathbf{V}}_k(\tilde{\theta}_k - \theta_\star)$$

$$= (\tilde{\theta}_k - \theta_\star)^{\top}\tilde{\mathbf{V}}_k(\tilde{\mathbf{V}}_k^{-1}\mathbf{P}_{\mathcal{S}_k}\mathbf{\Phi}_{\mathcal{D}_k}^{\top}\mathbf{y}_{\mathcal{D}_k} - \theta_\star)$$

$$= (\tilde{\theta}_k - \theta_\star)^{\top}\tilde{\mathbf{V}}_k[\tilde{\mathbf{V}}_k^{-1}\mathbf{P}_{\mathcal{S}_k}\mathbf{\Phi}_{\mathcal{D}_k}^{\top}(\mathbf{\Phi}_{\mathcal{D}_k}\theta_\star + \eta_{\mathcal{D}_k}) - \theta_\star]$$

$$= \underbrace{(\tilde{\theta}_k - \theta_\star)^{\top}\tilde{\mathbf{V}}_k(\tilde{\mathbf{V}}_k^{-1}\mathbf{P}_{\mathcal{S}_k}\mathbf{\Phi}_{\mathcal{D}_k}^{\top}\mathbf{\Phi}_{\mathcal{D}_k}\theta_\star - \theta_\star)}_{A_1} + \underbrace{(\tilde{\theta}_k - \theta_\star)^{\top}\mathbf{P}_{\mathcal{S}_k}\mathbf{\Phi}_{\mathcal{D}_k}^{\top}\eta_{\mathcal{D}_k}}_{A_2}$$

Since $\tilde{\mathbf{V}}_k(\tilde{\mathbf{V}}_k^{-1}\mathbf{P}_{\mathcal{S}_k}\mathbf{\Phi}_{\mathcal{D}_k}^{\top}\mathbf{\Phi}_{\mathcal{D}_k}\theta_\star - \theta_\star) = \mathbf{P}_{\mathcal{S}_k}\mathbf{\Phi}_{\mathcal{D}_k}^{\top}\mathbf{\Phi}_{\mathcal{D}_k}\theta_\star - \mathbf{P}_{\mathcal{S}_k}\mathbf{\Phi}_{\mathcal{D}_k}^{\top}\mathbf{\Phi}_{\mathcal{D}_k}\mathbf{P}_{\mathcal{S}_k}\theta_\star - \lambda\theta_\star = \mathbf{P}_{\mathcal{S}_k}\mathbf{\Phi}_{\mathcal{D}_k}^{\top}\mathbf{\Phi}_{\mathcal{D}_k}(\mathbf{I} - \mathbf{P}_{\mathcal{S}_k})\theta_\star - \lambda\theta_\star$, we have

$$A_1 = (\tilde{\theta}_k - \theta_\star)^{\top}\mathbf{P}_{\mathcal{S}_k}\mathbf{\Phi}_{\mathcal{D}_k}^{\top}\mathbf{\Phi}_{\mathcal{D}_k}(\mathbf{I} - \mathbf{P}_{\mathcal{S}_k})\theta_\star - \lambda(\tilde{\theta}_k - \theta_\star)^{\top}\theta_\star$$

$$= (\tilde{\theta}_k - \theta_\star)^{\top}\tilde{\mathbf{V}}_k^{1/2}\tilde{\mathbf{V}}_k^{-1/2}\mathbf{P}_{\mathcal{S}_k}\mathbf{\Phi}_{\mathcal{D}_k}^{\top}\mathbf{\Phi}_{\mathcal{D}_k}(\mathbf{I} - \mathbf{P}_{\mathcal{S}_k})\theta_\star - \lambda(\tilde{\theta}_k - \theta_\star)^{\top}\tilde{\mathbf{V}}_k^{1/2}\tilde{\mathbf{V}}_k^{-1/2}\theta_\star$$

$$\leq \|\tilde{\theta}_k - \theta_\star\|_{\tilde{\mathbf{V}}_k}\big(\|\tilde{\mathbf{V}}_k^{-1/2}\mathbf{P}_{\mathcal{S}_k}\mathbf{\Phi}_{\mathcal{D}_k}^{\top}\mathbf{\Phi}_{\mathcal{D}_k}(\mathbf{I} - \mathbf{P}_{\mathcal{S}_k})\theta_\star\| + \lambda\|\theta_\star\|_{\tilde{\mathbf{V}}_k^{-1}}\big)$$

$$\leq \|\tilde{\theta}_k - \theta_\star\|_{\tilde{\mathbf{V}}_k}\big(\|\tilde{\mathbf{V}}_k^{-1/2}\mathbf{P}_{\mathcal{S}_k}\mathbf{\Phi}_{\mathcal{D}_k}^{\top}\|\|\mathbf{\Phi}_{\mathcal{D}_k}(\mathbf{I} - \mathbf{P}_{\mathcal{S}_k})\|\|\theta_\star\| + \sqrt{\lambda}\|\theta_\star\|\big)$$

$$\leq \|\tilde{\theta}_k - \theta_\star\|_{\tilde{\mathbf{V}}_k}\big(\|\mathbf{\Phi}_{\mathcal{D}_k}(\mathbf{I} - \mathbf{P}_{\mathcal{S}_k})\| + \sqrt{\lambda}\big)\|\theta_\star\|$$

where the first inequality is due to Cauchy Schwartz, and the last inequality is because $\|\tilde{\mathbf{V}}_k^{-1/2}\mathbf{P}_{\mathcal{S}_k}\mathbf{\Phi}_{\mathcal{D}_k}^\top\| = \sqrt{\mathbf{\Phi}_{\mathcal{D}_k}\mathbf{P}_{\mathcal{S}_k}(\mathbf{P}_{\mathcal{S}_k}\mathbf{\Phi}_{\mathcal{D}_k}^\top\mathbf{\Phi}_{\mathcal{D}_k}\mathbf{P}_{\mathcal{S}_k} + \lambda\mathbf{I})^{-1}\mathbf{P}_{\mathcal{S}_k}\mathbf{\Phi}_{\mathcal{D}_k}^\top} \leq 1$. Then by definition of the spectral norm $\|\cdot\|$, and the properties of the orthogonal projection matrix $\mathbf{P}_{\mathcal{S}_k}$, we have

$$\|\mathbf{\Phi}_{\mathcal{D}_k}(\mathbf{I} - \mathbf{P}_{\mathcal{S}_k})\| = \sqrt{\lambda_{\max}\left(\mathbf{\Phi}_{\mathcal{D}_k}(\mathbf{I} - \mathbf{P}_{\mathcal{S}_k})^2\mathbf{\Phi}_{\mathcal{D}_k}^\top\right)} = \sqrt{\lambda_{\max}\left(\mathbf{\Phi}_{\mathcal{D}_k}(\mathbf{I} - \mathbf{P}_{\mathcal{S}_k})\mathbf{\Phi}_{\mathcal{D}_k}^\top\right)}.$$

Moreover, due to Lemma 4.1, $\mathcal{S}_k$ is $\epsilon$-accurate w.r.t. $\mathcal{D}_k$, for all $k$, so we have $\mathbf{I} - \mathbf{P}_{\mathcal{S}_k} \preceq \frac{\lambda}{1-\epsilon}(\mathbf{\Phi}_{\mathcal{D}_k}^\top\mathbf{\Phi}_{\mathcal{D}_k} + \lambda\mathbf{I})^{-1}$ by the property of $\epsilon$-accuracy (Proposition 10 of Calandriello et al. (2019)). Substituting this to RHS of the equality above, we have

$$\|\mathbf{\Phi}_{\mathcal{D}_k}(\mathbf{I} - \mathbf{P}_{\mathcal{S}_k})\| \leq \sqrt{\frac{\lambda}{1-\epsilon}\lambda_{\max}\left(\mathbf{\Phi}_{\mathcal{D}_k}(\mathbf{\Phi}_{\mathcal{D}_k}^\top\mathbf{\Phi}_{\mathcal{D}_k} + \lambda\mathbf{I})^{-1}\mathbf{\Phi}_{\mathcal{D}_k}^\top\right)} \leq \sqrt{\frac{\lambda}{1-\epsilon}}.$$

Therefore, $A_1 \leq \|\tilde{\theta}_k - \theta_\star\|_{\tilde{\mathbf{V}}_k}\left(\sqrt{\frac{1}{1-\epsilon}} + 1\right)\sqrt{\lambda}\|\theta_\star\|$.

Similarly, by applying Cauchy-Schwartz inequality on term $A_2$, we have

$$\begin{aligned}
A_2 &= (\tilde{\theta}_k - \theta_\star)^\top\tilde{\mathbf{V}}_k^{1/2}\tilde{\mathbf{V}}_k^{-1/2}\mathbf{P}_{\mathcal{S}_k}\mathbf{\Phi}_{\mathcal{D}_k}^\top\eta_{\mathcal{D}_k} \leq \|\tilde{\theta}_k - \theta_\star\|_{\tilde{\mathbf{V}}_k}\|\tilde{\mathbf{V}}_k^{-1/2}\mathbf{P}_{\mathcal{S}_k}\mathbf{\Phi}_{\mathcal{D}_k}^\top\eta_{\mathcal{D}_k}\| \\
&= \|\tilde{\theta}_k - \theta_\star\|_{\tilde{\mathbf{V}}_k}\|\tilde{\mathbf{V}}_k^{-1/2}\mathbf{P}_{\mathcal{S}_k}\mathbf{V}_k^{1/2}\mathbf{V}_k^{-1/2}\mathbf{\Phi}_{\mathcal{D}_k}^\top\eta_{\mathcal{D}_k}\| \\
&\leq \|\tilde{\theta}_k - \theta_\star\|_{\tilde{\mathbf{V}}_k}\|\tilde{\mathbf{V}}_k^{-1/2}\mathbf{P}_{\mathcal{S}_k}\mathbf{V}_k^{1/2}\|\|\mathbf{V}_k^{-1/2}\mathbf{\Phi}_{\mathcal{D}_k}^\top\eta_{\mathcal{D}_k}\|
\end{aligned}$$

where $\mathbf{V}_k := \mathbf{\Phi}_{\mathcal{D}_k}^\top\mathbf{\Phi}_{\mathcal{D}_k} + \lambda\mathbf{I}$. Note that $\mathbf{P}_{\mathcal{S}_k}\mathbf{V}_{k_k}\mathbf{P}_{\mathcal{S}_k} = \mathbf{P}_{\mathcal{S}_k}(\mathbf{\Phi}_{\mathcal{D}_k}^\top\mathbf{\Phi}_{\mathcal{D}_k} + \lambda\mathbf{I})\mathbf{P}_{\mathcal{S}_k} = \tilde{\mathbf{V}}_k + \lambda(\mathbf{P}_{\mathcal{S}_k} - \mathbf{I})$ and $\mathbf{P}_{\mathcal{S}_k} \preceq \mathbf{I}$, so we have

$$\begin{aligned}
\|\tilde{\mathbf{V}}_k^{-1/2}\mathbf{P}_{\mathcal{S}_k}\mathbf{V}_k^{1/2}\| &= \sqrt{\|\tilde{\mathbf{V}}_k^{-1/2}\mathbf{P}_{\mathcal{S}_k}\mathbf{V}_k^{1/2}\mathbf{V}_k^{1/2}\mathbf{P}_{\mathcal{S}_k}\tilde{\mathbf{V}}_k^{-1/2}\|} \leq \sqrt{\|\tilde{\mathbf{V}}_k^{-1/2}(\tilde{\mathbf{V}}_k + \lambda(\mathbf{P}_{\mathcal{S}_k} - \mathbf{I}))\tilde{\mathbf{V}}_k^{-1/2}\|} \\
&= \sqrt{\|\mathbf{I} + \lambda\tilde{\mathbf{V}}_k^{-1/2}(\mathbf{P}_{\mathcal{S}_k} - \mathbf{I}))\tilde{\mathbf{V}}_k^{-1/2}\|} \leq \sqrt{1 + \lambda\|\tilde{\mathbf{V}}_k^{-1}\|\|\mathbf{P}_{\mathcal{S}_k} - \mathbf{I}\|} \\
&\leq \sqrt{1 + \lambda \cdot \lambda^{-1} \cdot 1} = \sqrt{2},
\end{aligned}$$

and thus $A_2 \leq \sqrt{2}\|\tilde{\theta}_k - \theta_\star\|_{\tilde{\mathbf{V}}_k}\|\mathbf{V}_k^{-1/2}\mathbf{\Phi}_{\mathcal{D}_k}^\top\eta_{\mathcal{D}_k}\|$.

As mentioned by He et al. (2022), the standard self-normalized bound for vector-valued martingales cannot be directly applied to bound the term $\|\mathbf{V}_k^{-1/2}\mathbf{\Phi}_{\mathcal{D}_k}^\top\eta_{\mathcal{D}_k}\|$, since $\mathcal{D}_k$ is constructed by the data that each client has uploaded so far during the event-triggered communications. Therefore, in the following paragraphs, we bound this term by extending their results to the kernel bandit problem considered in our paper.

We first need to establish the following lemma.

**Lemma C.1.** *Let's denote* $\mathbf{V}_k(i) = \sum_{s\in\mathcal{N}_{t_{\underline{k}(i)}}(i)}\phi(\mathbf{x}_s)\phi(\mathbf{x}_s)^\top$, *such that* $\mathbf{V}_k = \lambda\mathbf{I} + \sum_{i=1}^N\mathbf{V}_k(i)$, *and then denote the covariance matrix for client $i$'s data that hasn't been uploaded to server by time step $t_k$ as* $\Delta\mathbf{V}_k(i) = \sum_{s\in\mathcal{N}_{t_k}(i)\backslash\mathcal{N}_{t_{\underline{k}(i)}}(i)}\phi(\mathbf{x}_s)\phi(\mathbf{x}_s)^\top$ *for $i \in [N]$. Then we have*

$$\mathbf{V}_k \succeq \frac{1}{\beta D}\Delta\mathbf{V}_k(i), \tag{8}$$

*and $\forall\mathbf{x} \in \mathbb{R}^d$,*

$$\frac{\phi(\mathbf{x})^\top\mathbf{V}_k^{-1}\phi(\mathbf{x})}{\phi(\mathbf{x})^\top(\mathbf{\Phi}_{[t_k]}^\top\mathbf{\Phi}_{[t_k]} + \lambda\mathbf{I})^{-1}\phi(\mathbf{x})} \leq 1 + N\beta D.$$

*Bounding* $\|\mathbf{V}_k^{-1/2}\mathbf{\Phi}_{\mathcal{D}_k}^\top\eta_{\mathcal{D}_k}\|$: Recall that $\mathcal{D}_k$ contains data points that $N$ clients have uploaded up to the $k$-th communication, i.e., $\mathcal{D}_k = \cup_{i=1}^N\mathcal{N}_{t_{\underline{k}(i)}}(i)$, where $t_{\underline{k}(i)}$ denotes the time step of client $i$'s

last communication with the server. Therefore, we have the following decomposition

$$
\begin{aligned}
\mathbf{V}_k^{-1/2}\boldsymbol{\Phi}_{\mathcal{D}_k}^\top \eta_{\mathcal{D}_k} &= \sum_{i=1}^N \mathbf{V}_k^{-1/2}\boldsymbol{\Phi}_{\mathcal{N}_{t_{\underline{k}(i)}}(i)}^\top \eta_{\mathcal{N}_{t_{\underline{k}(i)}}(i)} \\
&= \sum_{i=1}^N \mathbf{V}_k^{-1/2}\Big[\boldsymbol{\Phi}_{\mathcal{N}_{t_{\underline{k}(i)}}(i)}^\top \eta_{\mathcal{N}_{t_{\underline{k}(i)}}(i)} + \boldsymbol{\Phi}_{\mathcal{N}_{t_k}(i)\backslash\mathcal{N}_{t_{\underline{k}(i)}}(i)}^\top \eta_{\mathcal{N}_{t_k}(i)\backslash\mathcal{N}_{t_{\underline{k}(i)}}(i)}\Big] \\
&\quad - \sum_{i=1}^N \mathbf{V}_k^{-1/2}\boldsymbol{\Phi}_{\mathcal{N}_{t_k}(i)\backslash\mathcal{N}_{t_{\underline{k}(i)}}(i)}^\top \eta_{\mathcal{N}_{t_k}(i)\backslash\mathcal{N}_{t_{\underline{k}(i)}}(i)} \\
&= \mathbf{V}_k^{-1/2}\boldsymbol{\Phi}_{[t_k]}^\top \eta_{[t_k]} - \sum_{i=1}^N \mathbf{V}_k^{-1/2}\boldsymbol{\Phi}_{\mathcal{N}_{t_k}(i)\backslash\mathcal{N}_{t_{\underline{k}(i)}}(i)}^\top \eta_{\mathcal{N}_{t_k}(i)\backslash\mathcal{N}_{t_{\underline{k}(i)}}(i)}.
\end{aligned}
$$

Then using triangle inequality, we have

$$
\|\mathbf{V}_k^{-1/2}\boldsymbol{\Phi}_{\mathcal{D}_k}^\top \eta_{\mathcal{D}_k}\| \le \|\mathbf{V}_k^{-1/2}\boldsymbol{\Phi}_{[t_k]}^\top \eta_{[t_k]}\| + \sum_{i=1}^N \|\mathbf{V}_k^{-1/2}\boldsymbol{\Phi}_{\mathcal{N}_{t_k}(i)\backslash\mathcal{N}_{t_{\underline{k}(i)}}(i)}^\top \eta_{\mathcal{N}_{t_k}(i)\backslash\mathcal{N}_{t_{\underline{k}(i)}}(i)}\|.
$$

We can bound $\|\mathbf{V}_k^{-1/2}\boldsymbol{\Phi}_{[t_k]}^\top \eta_{[t_k]}\|$ as

$$
\begin{aligned}
\|\mathbf{V}_k^{-1/2}\boldsymbol{\Phi}_{[t_k]}^\top \eta_{[t_k]}\| &= \|\boldsymbol{\Phi}_{[t_k]}^\top \eta_{[t_k]}\|_{\mathbf{V}_k^{-1}} \le \|\boldsymbol{\Phi}_{[t_k]}^\top \eta_{[t_k]}\|_{(\boldsymbol{\Phi}_{[t_k]}^\top \boldsymbol{\Phi}_{[t_k]}+\lambda\mathbf{I})^{-1}}\sqrt{1+ND\beta} \\
&\le \sqrt{1+ND\beta}R\sqrt{2\ln(1/\delta)+\ln(\det(\mathbf{K}_{[T],[T]}/\lambda + \mathbf{I}))},
\end{aligned}
$$

with probability at least $1-\delta$, where the first inequality is due to Lemma C.1, and the second inequality is due to the standard self-normalized bound for kernelized contextual bandit, e.g., Lemma B.3. of Li et al. (2022).

Then we can bound $\|\mathbf{V}_k^{-1/2}\boldsymbol{\Phi}_{\mathcal{N}_{t_k}(i)\backslash\mathcal{N}_{t_{\underline{k}(i)}}(i)}^\top \eta_{\mathcal{N}_{t_k}(i)\backslash\mathcal{N}_{t_{\underline{k}(i)}}(i)}\|$ as

$$
\begin{aligned}
&\|\mathbf{V}_k^{-1/2}\boldsymbol{\Phi}_{\mathcal{N}_{t_k}(i)\backslash\mathcal{N}_{t_{\underline{k}(i)}}(i)}^\top \eta_{\mathcal{N}_{t_k}(i)\backslash\mathcal{N}_{t_{\underline{k}(i)}}(i)}\| \\
&\le \sqrt{2D\beta}\big\|\big[D\beta\lambda\mathbf{I}+\boldsymbol{\Phi}_{\mathcal{N}_{t_k}(i)\backslash\mathcal{N}_{t_{\underline{k}(i)}}(i)}^\top \boldsymbol{\Phi}_{\mathcal{N}_{t_k}(i)\backslash\mathcal{N}_{t_{\underline{k}(i)}}(i)}\big]^{-1/2}\boldsymbol{\Phi}_{\mathcal{N}_{t_k}(i)\backslash\mathcal{N}_{t_{\underline{k}(i)}}(i)}^\top \eta_{\mathcal{N}_{t_k}(i)\backslash\mathcal{N}_{t_{\underline{k}(i)}}(i)}\big\| \\
&= \sqrt{2D\beta}\big\|\boldsymbol{\Phi}_{\mathcal{N}_{t_k}(i)\backslash\mathcal{N}_{t_{\underline{k}(i)}}(i)}^\top \eta_{\mathcal{N}_{t_k}(i)\backslash\mathcal{N}_{t_{\underline{k}(i)}}(i)}\big\|_{\big[D\beta\lambda\mathbf{I}+\boldsymbol{\Phi}_{\mathcal{N}_{t_k}(i)\backslash\mathcal{N}_{t_{\underline{k}(i)}}(i)}^\top \boldsymbol{\Phi}_{\mathcal{N}_{t_k}(i)\backslash\mathcal{N}_{t_{\underline{k}(i)}}(i)}\big]^{-1}} \\
&\le \sqrt{2D\beta}R\sqrt{2\ln(1/\delta)+\ln(\det(\mathbf{K}_{[T],[T]}/(D\beta\lambda)+\mathbf{I}))}
\end{aligned}
$$

where the first inequality is because $\mathbf{V}_k = \lambda\mathbf{I}+\boldsymbol{\Phi}_{\mathcal{D}_k}^\top\boldsymbol{\Phi}_{\mathcal{D}_k} \succeq \frac{1}{D\beta}\boldsymbol{\Phi}_{\mathcal{N}_{t_k}(i)\backslash\mathcal{N}_{t_{\underline{k}(i)}}(i)}^\top \boldsymbol{\Phi}_{\mathcal{N}_{t_k}(i)\backslash\mathcal{N}_{t_{\underline{k}(i)}}(i)}$ due to equation 8 in Lemma C.1, so $\mathbf{V}_k = \lambda\mathbf{I}+\boldsymbol{\Phi}_{\mathcal{D}_k}^\top\boldsymbol{\Phi}_{\mathcal{D}_k} \succeq \frac{1}{2D\beta}(D\beta\lambda\mathbf{I}+\boldsymbol{\Phi}_{\mathcal{N}_{t_k}(i)\backslash\mathcal{N}_{t_{\underline{k}(i)}}(i)}^\top \boldsymbol{\Phi}_{\mathcal{N}_{t_k}(i)\backslash\mathcal{N}_{t_{\underline{k}(i)}}(i)})$, and the second inequality is again obtained using the standard self-normalized bound.

Putting everything together, we have

$$
\|\tilde{\theta}_k - \theta_\star\|_{\tilde{\mathbf{V}}_k} \le (\sqrt{1/(1-\epsilon)}+1)\sqrt{\lambda}\|\theta_\star\| + 2\big(\sqrt{1+ND\beta}+N\sqrt{2D\beta}\big)R\sqrt{\ln(1/\delta)+\gamma_T},
$$

where $\gamma_T := \max_{\mathcal{D}\subset\mathcal{A}:|\mathcal{D}|=T}\frac{1}{2}\log\det(\mathbf{K}_{\mathcal{D},\mathcal{D}}/(D\beta\lambda)+\mathbf{I})$.

*Proof of Lemma C.1.* Note that by definition, $\Delta\mathbf{V}_k(c_k) = \mathbf{0}$, where $c_k \in [N]$ is the index of the client who triggers the $k$-th communication. In the following, we first show that

$$
\mathbf{V}_k \succeq \frac{1}{\beta D}\Delta\mathbf{V}_k(i)
$$

for all $i \in [N]$. For client $c_k$, $\mathbf{V}_k \succeq \mathbf{0} = \frac{1}{\beta D}\Delta\mathbf{V}_k(c_k)$. For client $i \neq c_k$, we have

$$\frac{\phi(\mathbf{x})^\top \mathbf{V}_{\underline{k}(i)}^{-1}\phi(\mathbf{x})}{\phi(\mathbf{x})^\top \left(\mathbf{V}_{\underline{k}(i)} + \Delta\mathbf{V}_k(i)\right)^{-1}\phi(\mathbf{x})}$$

$$\leq 1 + \sum_{s \in \mathcal{N}_{t_k}(i)\backslash\mathcal{N}_{t_{\underline{k}(i)}}(i)} \phi(\mathbf{x}_s)^\top \mathbf{V}_{\underline{k}(i)}^{-1}\phi(\mathbf{x}_s) = 1 + \sum_{s \in \mathcal{N}_{t_k}(i)\backslash\mathcal{N}_{t_{\underline{k}(i)}}(i)} \sigma_{\underline{k}(i)}^2(\mathbf{x}_s)$$

$$\leq 1 + \beta \sum_{s \in \mathcal{N}_{t_k}(i)\backslash\mathcal{N}_{t_{\underline{k}(i)}}(i)} \tilde{\sigma}_{\underline{k}(i)}^2(\mathbf{x}_s) \leq 1 + \beta D,$$

where the first inequality is due to Lemma A.5, the second is due to property of $\epsilon$-accuracy in Lemma A.6, and the third is due to our event-trigger in equation 4.

This implies $\mathbf{V}_{\underline{k}(i)}^{-1} \preceq (1 + \beta D)\left(\mathbf{V}_{\underline{k}(i)} + \Delta\mathbf{V}_k(i)\right)^{-1}$. Then due to Lemma A.9, we have $(1 + \beta D)\mathbf{V}_{\underline{k}(i)} \succeq \mathbf{V}_{\underline{k}(i)} + \Delta\mathbf{V}_k(i)$, and thus $\mathbf{V}_{\underline{k}(i)} \succeq \frac{1}{\beta D}\Delta\mathbf{V}_k(i)$. In addition, since $\underline{k}(i) < k, \forall i \neq c_k$, we have $\mathbf{V}_k \succeq \mathbf{V}_{\underline{k}(i)} \succeq \frac{1}{\beta D}\Delta\mathbf{V}_k(i)$.

By averaging equation 8 over all $N$ clients, we have

$$\mathbf{V}_k \succeq \frac{1}{N\beta D}\sum_{i=1}^{N}\Delta\mathbf{V}_k(i),$$

and thus, we have

$$\mathbf{\Phi}_{[t_k]}^\top \mathbf{\Phi}_{[t_k]} + \lambda\mathbf{I} = \mathbf{V}_k + \sum_{i=1}^{N}\Delta\mathbf{V}_k(i) \preceq (1 + N\beta D)\mathbf{V}_k.$$

Using Lemma A.9 again finishes the proof. $\qquad\square$

## D  PROOF OF THEOREM 4.5 IN SECTION 4.3

### D.1  COMMUNICATION COST

Recall from Section 4.1 that $\mathcal{D}_k$ is the set of time indices for the data points that are used to construct the embedded statistics on the server at the $k$-th communication round, for $k = 1, \dots, B$. We denote the corresponding (exact) covariance matrix as $\mathbf{V}_k = \lambda\mathbf{I} + \mathbf{\Phi}_{\mathcal{D}_k}^\top\mathbf{\Phi}_{\mathcal{D}_k} \in \mathbb{R}^{p \times p}$, with $\mathbf{V}_0 = \lambda\mathbf{I}$, and kernel matrix as $\mathbf{K}_{\mathcal{D}_k,\mathcal{D}_k} = \mathbf{\Phi}_{\mathcal{D}_k}\mathbf{\Phi}_{\mathcal{D}_k}^\top \in \mathbb{R}^{|\mathcal{D}_k| \times |\mathcal{D}_k|}$.

Similar to (He et al., 2022), by defining $k_p = \min\{k \in [B] \mid \det(\mathbf{I} + \lambda^{-1}\mathbf{K}_{\mathcal{D}_k,\mathcal{D}_k}) \geq 2^p)\}$, we have $\log\left(\det(\mathbf{I} + \lambda^{-1}\mathbf{K}_{\mathcal{D}_{k_{p+1}},\mathcal{D}_{k_{p+1}}})/\det(\mathbf{I} + \lambda^{-1}\mathbf{K}_{\mathcal{D}_{k_p},\mathcal{D}_{k_p}})\right) \geq 1$ for each $p \geq 0$. We call the sequence of time steps in-between $t_{k_p}$ and $t_{k_{p+1}}$ an epoch, and denote the total number of epochs as $P$. Note that since

$$\log\left(\frac{\det(\mathbf{I} + \lambda^{-1}\mathbf{K}_{\mathcal{D}_{k_1},\mathcal{D}_{k_1}})}{\det(\mathbf{I})}\right) + \log\left(\frac{\det(\mathbf{I} + \lambda^{-1}\mathbf{K}_{\mathcal{D}_{k_2},\mathcal{D}_{k_2}})}{\det(\mathbf{I} + \lambda^{-1}\mathbf{K}_{\mathcal{D}_{k_1},\mathcal{D}_{k_1}})}\right) + \cdots + \log\left(\frac{\det(\mathbf{I} + \lambda^{-1}\mathbf{K}_{\mathcal{D}_{k_P},\mathcal{D}_{k_P}})}{\det(\mathbf{I} + \lambda^{-1}\mathbf{K}_{\mathcal{D}_{k_{P-1}},\mathcal{D}_{k_{P-1}}})}\right)$$

$$\leq \log\left(\det(\mathbf{I} + \lambda^{-1}\mathbf{K}_{[T],[T]})\right) \leq 2\gamma_T,$$

there can be at most $2\gamma_T$ terms, i.e., $P \leq 2\gamma_T$. Now that we have divided the time horizon $[T]$ into $P$ epochs using $\{t_{k_p}\}_{p \in [P]}$, we prove the following lemma that upper bounds the total number of times communication is triggered in each epoch.

**Lemma D.1** (Number of communications per epoch). *For each epoch, i.e., the sequence of time steps in-between $t_{k_p}$ and $t_{k_{p+1}}$, the number of communications is upper bounded by $N + \frac{4\beta}{D}$.*

Since there are at most $2\gamma_T$ epochs, the total number of communications $B \leq 2\gamma_T(N + \frac{4\beta}{D})$. Moreover, by Lemma 4.1, we know that during each communication, the size of data being communicated is $O\left(\log^2(T)\gamma_T^2\right)$. Hence, with $D = \frac{1}{N^2}$, $C_T = O(N^2\gamma_T^3\log^2(T))$.

*Proof of Lemma D.1.* Consider the epoch $[t_{k_p}, t_{k_{p+1}} - 1]$ for some $p = 0, 1, \ldots, P$. We denote the total number of communications in this epoch as $Q_p$, and the total number of communications in this epoch that are triggered by client $i$ as $Q_{p,i}$ for $i \in [N]$, i.e., $Q_p = \sum_{i=1}^{N} Q_{p,i}$.

Let's denote the indices associated with the communications triggered by some client $i$ as $\kappa_1, \kappa_2, \ldots, \kappa_{Q_{p,i}} \in [k_p, k_{p+1} - 1]$. Then for each $j = 2, 3, \ldots, Q_{p,i}$, i.e., excluding client $i$'s first communication in this epoch, due to our event-trigger design in equation 4, we have

$$\beta \sum_{s \in \Delta\mathcal{D}_{\kappa_j}} \sigma^2_{k_p}(\mathbf{x}_s) \geq \beta \sum_{s \in \Delta\mathcal{D}_{\kappa_j}} \sigma^2_{\kappa_{j-1}}(\mathbf{x}_s) \geq \sum_{s \in \Delta\mathcal{D}_{\kappa_j}} \tilde{\sigma}^2_{\kappa_{j-1}}(\mathbf{x}_s) > D,$$

where the first inequality is because by definition of $\kappa_{j-1}$, we have $\kappa_{j-1} \geq k_p$, so $\sigma^2_{\kappa_{j-1}}(\mathbf{x}) \leq \sigma^2_{k_p}(\mathbf{x}), \forall \mathbf{x}$, and the second inequality is due to Lemma A.6. Therefore, we have $\sum_{s \in \Delta\mathcal{D}_{k_j}} \sigma^2_{k_p}(\mathbf{x}_s) \geq D/\beta$. Since $\sigma^2_{k_p}(\mathbf{x}) = \|\phi(\mathbf{x})\|^2_{\mathbf{V}^{-1}_{k_p}}$, we have

$$D/\beta \leq \sum_{s \in \Delta\mathcal{D}_{k_j}} \|\phi(\mathbf{x}_s)\|^2_{\mathbf{V}^{-1}_{k_p}} \leq 4\log\Big(\frac{\det(\mathbf{I} + \lambda^{-1}\mathbf{K}_{\mathcal{D}_{k_p} \cup \Delta\mathcal{D}_{\kappa_j}, \mathcal{D}_{k_p} \cup \Delta\mathcal{D}_{\kappa_j}})}{\det(\mathbf{I} + \lambda^{-1}\mathbf{K}_{\mathcal{D}_{k_p}, \mathcal{D}_{k_p}})}\Big)$$

$$\leq -4 + 4\frac{\det(\mathbf{I} + \lambda^{-1}\mathbf{K}_{\mathcal{D}_{k_p} \cup \Delta\mathcal{D}_{\kappa_j}, \mathcal{D}_{k_p} \cup \Delta\mathcal{D}_{\kappa_j}})}{\det(\mathbf{I} + \lambda^{-1}\mathbf{K}_{\mathcal{D}_{k_p}, \mathcal{D}_{k_p}})}$$

where the second inequality is by definition of epoch, i.e., $\det(\mathbf{I} + \lambda^{-1}\mathbf{K}_{\mathcal{D}_{k_{p+1}-1}, \mathcal{D}_{k_{p+1}-1}})/\det(\mathbf{I} + \lambda^{-1}\mathbf{K}_{\mathcal{D}_{k_p}, \mathcal{D}_{k_p}}) \leq 2$, combined with Lemma A.4, and the third is because $\log(x) \leq x - 1$ for $x > 0$. Hence, we have

$$\frac{\det(\mathbf{I} + \lambda^{-1}\mathbf{K}_{\mathcal{D}_{k_p} \cup \Delta\mathcal{D}_{\kappa_j}, \mathcal{D}_{k_p} \cup \Delta\mathcal{D}_{\kappa_j}})}{\det(\mathbf{I} + \lambda^{-1}\mathbf{K}_{\mathcal{D}_{k_p}, \mathcal{D}_{k_p}})} \geq 1 + \frac{D}{4\beta},$$

and thus, we have $\det(\mathbf{I} + \lambda^{-1}\mathbf{K}_{\mathcal{D}_{k_p} \cup \Delta\mathcal{D}_{\kappa_j}, \mathcal{D}_{k_p} \cup \Delta\mathcal{D}_{\kappa_j}}) - \det(\mathbf{I} + \lambda^{-1}\mathbf{K}_{\mathcal{D}_{k_p}, \mathcal{D}_{k_p}}) \geq \frac{D}{4\beta}\det(\mathbf{I} + \lambda^{-1}\mathbf{K}_{\mathcal{D}_{k_p}, \mathcal{D}_{k_p}})$ for all $j = 2, 3, \ldots, Q_{p,i}$ and all client $i \in [N]$.

Denote the indices associated with the communications of all clients in this epoch as $\kappa'_1, \kappa'_2, \ldots, \kappa'_{Q_p} \in \{k_p, k_{p+1} - 1\}$. Then for each $j \in [Q_p]$, if client $c_{\kappa'_j}$ has already communicated with the server ealier in this epoch, we have

$$\det(\mathbf{I} + \lambda^{-1}\mathbf{K}_{\mathcal{D}_{\kappa'_j}, \mathcal{D}_{\kappa'_j}}) - \det(\mathbf{I} + \lambda^{-1}\mathbf{K}_{\mathcal{D}_{\kappa'_{j-1}}, \mathcal{D}_{\kappa'_{j-1}}})$$

$$= \det(\mathbf{I} + \lambda^{-1}\mathbf{K}_{\mathcal{D}_{\kappa'_{j-1}} \cup \Delta\mathcal{D}_{\kappa'_j}, \mathcal{D}_{\kappa'_{j-1}} \cup \Delta\mathcal{D}_{\kappa'_j}}) - \det(\mathbf{I} + \lambda^{-1}\mathbf{K}_{\mathcal{D}_{\kappa'_{j-1}}, \mathcal{D}_{\kappa'_{j-1}}})$$

$$\geq \det(\mathbf{I} + \lambda^{-1}\mathbf{K}_{\mathcal{D}_{k_p} \cup \Delta\mathcal{D}_{\kappa'_j}, \mathcal{D}_{k_p} \cup \Delta\mathcal{D}_{\kappa'_j}}) - \det(\mathbf{I} + \lambda^{-1}\mathbf{K}_{\mathcal{D}_{k_p}, \mathcal{D}_{k_p}})$$

$$\geq \frac{D}{4\beta}\det(\mathbf{I} + \lambda^{-1}\mathbf{K}_{\mathcal{D}_{k_p}, \mathcal{D}_{k_p}})$$

where the first inequality is obtained via matrix determinant lemma and Lemma A.10, and the second is due to the inequality we derived above. Summing over all communications in this epoch, we have

$$\det(\mathbf{I} + \lambda^{-1}\mathbf{K}_{\mathcal{D}_{k_{p+1}-1}, \mathcal{D}_{k_{p+1}-1}}) - \det(\mathbf{I} + \lambda^{-1}\mathbf{K}_{\mathcal{D}_{k_p}, \mathcal{D}_{k_p}})$$

$$= \sum_{j=1}^{Q_p} \det(\mathbf{I} + \lambda^{-1}\mathbf{K}_{\mathcal{D}_{\kappa'_j}, \mathcal{D}_{\kappa'_j}}) - \det(\mathbf{I} + \lambda^{-1}\mathbf{K}_{\mathcal{D}_{\kappa'_{j-1}}, \mathcal{D}_{\kappa'_{j-1}}})$$

$$\geq \sum_{i=1}^{N} (Q_{p,i} - 1)\frac{D}{4\beta}\det(\mathbf{I} + \lambda^{-1}\mathbf{K}_{\mathcal{D}_{k_p}, \mathcal{D}_{k_p}}),$$

and since $\det(\mathbf{I} + \lambda^{-1}\mathbf{K}_{\mathcal{D}_{k_{p+1}-1}, \mathcal{D}_{k_{p+1}-1}})/\det(\mathbf{I} + \lambda^{-1}\mathbf{K}_{\mathcal{D}_{k_p}, \mathcal{D}_{k_p}}) \leq 2$ by our definition of epoch, we have

$$1 + \frac{D}{4\beta}\sum_{i=1}^{N}(Q_{p,i} - 1) \leq \det(\mathbf{I} + \lambda^{-1}\mathbf{K}_{\mathcal{D}_{k_{p+1}-1}, \mathcal{D}_{k_{p+1}-1}})/\det(\mathbf{I} + \lambda^{-1}\mathbf{K}_{\mathcal{D}_{k_p}, \mathcal{D}_{k_p}}) \leq 2,$$

so $Q_p = \sum_{i=1}^{N} Q_{p,i} \leq N + \frac{4\beta}{D}$, which finishes the proof. $\qquad\square$

## D.2 CUMULATIVE REGRET

To facilitate regret analysis of Async-KernelUCB, we need to introduce some additional notations. First, let's denote the client who triggers the $k$-th communication as $c_k \in [N]$, the index of its next communication as $\bar{k}(c_k)$, and the time step when the $\bar{k}(c_k)$-th communication happens is $t_{\bar{k}(c_k)}$ ($t_{\bar{k}(c_k)} = T$ if $k$ is client $c_k$'s final communication with the server). Then we denote the set of time steps in-between (but not including) the current (the $k$-th) communication and client $c_k$'s next communication when client $c_k$ is active as $\mathcal{P}_k := \{t_k < s < t_{\bar{k}(c_k)} : i_s = c_k\}$, and thus by definition $\Delta \mathcal{D}_{\bar{k}(c_k)} = \mathcal{N}_{t_{\bar{k}(c_k)}}(c_k) \setminus \mathcal{N}_{t_k}(c_k) = \mathcal{P}_k \cup \{t_{\bar{k}(c_k)}\}$. We also define $\mathcal{P}_0$ as the union over the set of time steps before the first communication of each client $i \in [N]$. Therefore, we have $(\cup_{k=0}^{B} \mathcal{P}_k) \cup \{t_k\}_{k \in [B]} = [T]$. Since in Algorithm 1, the approximated mean and variance of each client only get updated when it triggers communication, and then remain fixed until after its next communication, we have that all the interactions in $\mathcal{P}_k \cup \{t_{\bar{k}(c_k)}\}$ are based on the same $\{\tilde{\mu}_k(\cdot), \tilde{\sigma}_k(\cdot)\}$, for $k = 0, 1, \ldots, B$. In addition, an important observation is that, based on our event-trigger in equation 4, we have

$$\sum_{s \in \mathcal{P}_k} \tilde{\sigma}_k^2(\mathbf{x}_s) \leq D,$$

$$\Big[ \sum_{s \in \mathcal{P}_k} \tilde{\sigma}_k^2(\mathbf{x}_s) \Big] + \tilde{\sigma}_k^2(\mathbf{x}_{t_{\bar{k}(c_k)}}) > D. \tag{9}$$

Now we are ready to upper bound the cumulative regret. Consider some time step $t \in \mathcal{P}_k \cup \{t_{\bar{k}(c_k)}\}$. Due to our arm selection rule (line 5 of Algorithm 1), we have $\mathbf{x}_t = \arg\max_{\mathbf{x} \in \mathcal{A}_t} \tilde{\mu}_k(\mathbf{x}) + \alpha \tilde{\sigma}_k(\mathbf{x})$. Combining this with Lemma 4.4, with probability at least $1 - \delta$, we have

$$f(\mathbf{x}_t^\star) \leq \tilde{\mu}_k(\mathbf{x}_t^\star) + \alpha \tilde{\sigma}_k(\mathbf{x}_t^\star) \leq \tilde{\mu}_k(\mathbf{x}_t) + \alpha \tilde{\sigma}_k(\mathbf{x}_t),$$
$$f(\mathbf{x}_t) \geq \tilde{\mu}_k(\mathbf{x}_t) - \alpha \tilde{\sigma}_k(\mathbf{x}_t),$$

where $\mathbf{x}_t^\star := \arg\max_{\mathbf{x} \in \mathcal{A}_t} f(\mathbf{x}) = \arg\max_{\mathbf{x} \in \mathcal{A}_t} \phi(\mathbf{x})^\top \theta_\star$ is the optimal arm at time step $t$, and thus $r_t = f(\mathbf{x}_t^\star) - f(\mathbf{x}_t) \leq 2\alpha \tilde{\sigma}_k(\mathbf{x}_t)$. The cumulative regret $R_T$ can be rewritten as

$$R_T = \sum_{k=0}^{B} \sum_{s \in \mathcal{P}_k} r_s + \sum_{k=1}^{B} r_{t_k} \leq \sum_{k=0}^{B} \sum_{s \in \mathcal{P}_k} \min(2LS, 2\alpha \tilde{\sigma}_k(\mathbf{x}_s)) + \sum_{k=1}^{B} \min\{2LS, 2\alpha \tilde{\sigma}_{\underline{k}(c_k)}(\mathbf{x}_{t_k})\}.$$

*Bounding first term:* To bound the first term, we introduce an imaginary variance function $\sigma_{k,s-1}^2(\cdot)$ (not computed in the actual algorithm) for $s \in \mathcal{P}_k$ and $k = 0, 1, \ldots, B$, which is constructed using dataset $(\cup_{k'=0}^{k-1} \mathcal{P}_{k'}) \cup \{s' \in \mathcal{P}_k : s' \leq s - 1\}$. In the following paragraph, we will bound the first term by showing that $\sum_{k=0}^{B} \sum_{s \in \mathcal{P}_k} \tilde{\sigma}_k^2(\mathbf{x}_s)$ is not too much larger than $\sum_{k=0}^{B} \sum_{s \in \mathcal{P}_k} \sigma_{k,s-1}^2(\mathbf{x}_s)$.

This requires us to bound the ratio $\frac{\sigma_k^2(\mathbf{x}_s)}{\sigma_{k,s-1}^2(\mathbf{x}_s)}$ for $s \in \mathcal{P}_k$ and $k = 0, 1, \ldots, B$. Recall that $\sigma_k^2(\cdot)$ is constructed using data points that $N$ clients have uploaded to the server up to the $k$-th communication, i.e., $\mathcal{D}_k = \cup_{i=1}^{N} \mathcal{N}_{t_{\underline{k}(i)}}(t_k)$, which is a subset of $\mathcal{D}_k \cup (\cup_{i=1}^{N} \Delta \mathcal{D}_{\bar{k}(i)}) = \mathcal{D}_k \cup (\cup_{i=1}^{N} \mathcal{P}_{\underline{k}(i)} \cup \{t_{\bar{k}(i)}\})$. However, as shown in equation 9, the event-trigger cannot be directly used to upper bound the summation of approximated variances in $\mathcal{P}_{\underline{k}(i)} \cup \{t_{\bar{k}(i)}\}$, but can be used to upper bound that in $\mathcal{P}_{\underline{k}(i)}$, which is why we construct the imaginary variance function without using data points with time indices $\{t_k\}_{k \in [B]}$. Specifically, using the notations we just introduced, we can rewrite the variance as

$$\sigma_k^2(\mathbf{x}) = \phi(\mathbf{x})^\top \big( \mathbf{\Phi}_{\mathcal{D}_k}^\top \mathbf{\Phi}_{\mathcal{D}_k} + \lambda \mathbf{I} \big)^{-1} \phi(\mathbf{x})$$

$$\sigma_{k,s-1}^2(\mathbf{x}) = \phi(\mathbf{x})^\top \big( \mathbf{\Phi}_{\mathcal{D}_k \setminus \{t_{k'}\}_{k' \in [k]}}^\top \mathbf{\Phi}_{\mathcal{D}_k \setminus \{t_{k'}\}_{k' \in [k]}} + \lambda \mathbf{I} + \sum_{i \neq c_k} \mathbf{\Phi}_{\mathcal{P}_{\underline{k}(i)}}^\top \mathbf{\Phi}_{\mathcal{P}_{\underline{k}(i)}}$$

$$+ \mathbf{\Phi}_{\{s' \in \mathcal{P}_k : s' \leq s-1\}}^\top \mathbf{\Phi}_{\{s' \in \mathcal{P}_k : s' \leq s-1\}} \big)^{-1} \phi(\mathbf{x})$$

$$\geq \phi(\mathbf{x})^\top \big( \mathbf{\Phi}_{\mathcal{D}_k}^\top \mathbf{\Phi}_{\mathcal{D}_k} + \lambda \mathbf{I} + \sum_{i \neq c_k} \mathbf{\Phi}_{\mathcal{P}_{\underline{k}(i)}}^\top \mathbf{\Phi}_{\mathcal{P}_{\underline{k}(i)}} + \mathbf{\Phi}_{\{s' \in \mathcal{P}_k : s' \leq s-1\}}^\top \mathbf{\Phi}_{\{s' \in \mathcal{P}_k : s' \leq s-1\}} \big)^{-1} \phi(\mathbf{x})$$

The following lemma provides a upper bound for this ratio.

**Lemma D.2** (Bounding $\sigma_k^2(\mathbf{x}_s)/\sigma_{k,s-1}^2(\mathbf{x}_s)$). *Under the same condition as Lemma 4.1, with communication threshold $D$, we have $\forall k, s$ that*

$$\sigma_k^2(\mathbf{x}_s)/\sigma_{k,s-1}^2(\mathbf{x}_s) \leq 1 + N\beta D.$$

With Lemma D.2, we can bound the first term as

$$\sum_{k=0}^{B}\sum_{s\in\mathcal{P}_k}\min(2LS, 2\alpha\tilde{\sigma}_k(\mathbf{x}_s)) \leq 2\alpha\sqrt{T\sum_{k=0}^{B}\sum_{s\in\mathcal{P}_k}\tilde{\sigma}_k^2(\mathbf{x}_s)} \leq 2\alpha\sqrt{T\beta\sum_{k=0}^{B}\sum_{s\in\mathcal{P}_k}\sigma_k^2(\mathbf{x}_s)}$$

$$= 2\alpha\sqrt{T\beta\sum_{k=0}^{B}\sum_{s\in\mathcal{P}_k}\sigma_{k,s-1}^2(\mathbf{x}_s)\cdot\frac{\sigma_k^2(\mathbf{x}_s)}{\sigma_{k,s-1}^2(\mathbf{x}_s)}} \leq 2\alpha\sqrt{T\beta(1+N\beta D)\sum_{k=0}^{B}\sum_{s\in\mathcal{P}_k}\sigma_{k,s-1}^2(\mathbf{x}_s)}$$

$$\leq 4\alpha\sqrt{T\beta(1+N\beta D)\gamma_T}$$

$$\leq 4\Big[(1/\sqrt{1-\epsilon}+1)\sqrt{\lambda}S + 2R\big(\sqrt{1+ND\beta}+N\sqrt{2D\beta}\big)\sqrt{\ln(1/\delta)+\gamma_T}\Big]\sqrt{T\beta(1+N\beta D)\gamma_T}$$

with probability at least $1 - 2\delta$, where the first inequality is due to Cauchy-Schwarz, and second is due to the property of $\epsilon$-accuracy in Lemma A.6, the third is due to Lemma D.2, the forth is by definition of maximum information gain $\gamma_T$, and the last is by substituting $\alpha$ defined in Lemma 4.4.

*Bounding second term:* For the second term $\sum_{k=1}^{B}\min\{2LS, 2\alpha\tilde{\sigma}_{\underline{k}(c_k)}(\mathbf{x}_{t_k})\}$, we should note that $\tilde{\sigma}_{\underline{k}(c_k)}(\cdot)$ is the approximated variance function that client $c_k$ received during its last communication with the server, instead of $\sigma_{k-1}(\cdot)$ as in our proof of Lemma 4.3 when bounding the size of dictionary. Ideally, we want to relate each $\sigma_{\underline{k}(c_k)}(\cdot)$ to $\sigma_k(\cdot)$ and then apply the elliptical potential argument, but as we do not make any assumption on how frequent client arrives, it is possible that for clients who show up infrequently, these two functions are very different.

However, by using the epoch argument as in the proof for communication cost, we can show that this undesirable situation only occurs at most $2\gamma_T$ times. Specifically, recall that $\mathbf{V}_k = \lambda\mathbf{I} + \mathbf{\Phi}_{\mathcal{D}_k}^{\top}\mathbf{\Phi}_{\mathcal{D}_k}$, with $\mathbf{V}_0 = \lambda\mathbf{I}$, and kernel matrix as $\mathbf{K}_{\mathcal{D}_k,\mathcal{D}_k} = \mathbf{\Phi}_{\mathcal{D}_k}\mathbf{\Phi}_{\mathcal{D}_k}^{\top} \in \mathbb{R}^{|\mathcal{D}_k|\times|\mathcal{D}_k|}$. We define $k_p = \min\{k \in [B] \mid \det(\mathbf{I} + \lambda^{-1}\mathbf{K}_{\mathcal{D}_k,\mathcal{D}_k}) \geq 2^p)\}$, such that $\log\big(\det(\mathbf{I}+\lambda^{-1}\mathbf{K}_{\mathcal{D}_{k_{p+1}},\mathcal{D}_{k_{p+1}}})/\det(\mathbf{I}+\lambda^{-1}\mathbf{K}_{\mathcal{D}_{k_p},\mathcal{D}_{k_p}})\big) \geq 1$ for each $p \geq 0$. We call the sequence of time steps in-between $t_{k_p}$ and $t_{k_{p+1}}$ an epoch, and denote the total number of epochs as $P$. As shown in the proof for communication cost, we have $P \leq 2\gamma_T$.

Consider the epoch $[t_{k_p}, t_{k_{p+1}} - 1]$ for some $p = 0, 1, \ldots, P$. We denote the total number of communications in this epoch that are triggered by client $i$ as $Q_{p,i}$ for $i \in [N]$, and the indices associated with these communications triggered by client $i$ as $\kappa_1, \kappa_2, \ldots, \kappa_{Q_{p,i}} \in [k_p, k_{p+1} - 1]$.

As mentioned above, the approximated variance used during arm selection at $t_{\kappa_1}$, i.e, $\sigma_{\underline{\kappa_1}(c_{\kappa_1})}^2(\cdot)$ could be from a very long time ago. Therefore, we simply bound its regret by $2LS$, and in total, there can be at most $2\gamma_T N$ such terms for all $N$ clients, leading to a upper bound of $4N\gamma_T LS$.

Now we only need to be concerned about the communications at $j = 2, 3, \ldots, Q_{p,i}$, and show that $\sigma_{\underline{\kappa_j}(c_{\kappa_j})}^2(\mathbf{x})$ is close to $\sigma_{\kappa_j}^2(\mathbf{x})$ for all $\mathbf{x}$. Specifically, we have

$$\sigma_{\underline{\kappa_j}(c_{\kappa_j})}^2(\mathbf{x}) = \sigma_{\kappa_{j-1}}^2(\mathbf{x}) = \sigma_{\kappa_j}^2(\mathbf{x})\frac{\sigma_{\kappa_{j-1}}^2(\mathbf{x})}{\sigma_{\kappa_j}^2(\mathbf{x})} \leq 2\sigma_{\kappa_j}^2(\mathbf{x}),$$

where the first equality is because by definition $\underline{\kappa_j}(c_{\kappa_j}) = \kappa_{j-1}$, the first inequality is because $\sigma_{\kappa_{j-1}}^2(\mathbf{x})/\sigma_{\kappa_j}^2(\mathbf{x}) \leq \det(\mathbf{I} + \lambda^{-1}\mathbf{K}_{\mathcal{D}_{k_{p+1}-1},\mathcal{D}_{k_{p+1}-1}})/\det(\mathbf{I}+\lambda^{-1}\mathbf{K}_{\mathcal{D}_{k_p},\mathcal{D}_{k_p}}) \leq 2$ due to Lemma A.3, Lemma A.9 and the definition of epoch. Therefore, further applying Cauchy-Schwarz and the

$\epsilon$-accuracy property in Lemma A.6, the second term can be bounded by

$$\sum_{k=1}^{B} \min\{2LS, 2\alpha\tilde{\sigma}_{\underline{k}(c_k)}(\mathbf{x}_{t_k})\} \leq 4N\gamma_T LS + 2\alpha\sqrt{2B\beta \sum_{k=1}^{B} \sigma_k^2(\mathbf{x}_{t_k})}$$

$$\leq 4N\gamma_T LS + 2\alpha\sqrt{2B\beta \sum_{k=1}^{B} \sigma_{k-1,t_k-1}^2(\mathbf{x}_{t_k})} < 4N\gamma_T LS + 2\alpha\sqrt{2B\beta \sum_{k=1}^{B} \sum_{s \in \Delta\mathcal{D}_k} \sigma_{k-1,s-1}^2(\mathbf{x}_s)}$$

$$\leq 4N\gamma_T LS + 4\alpha\sqrt{2T\beta\gamma_T}$$

where the imaginary variance function $\sigma_{k-1,s-1}^2(\cdot)$ is constructed using dataset $\left(\cup_{k'=1}^{k-1}\Delta\mathcal{D}_{k'}\right) \cup \{s' \in \Delta\mathcal{D}_k : s' \leq s-1\}$, the second inequality is because variance is non-increasing over time, the third is because variances are positive, and the last is due to definition of maximum information gain $\gamma_T$ and that $B \leq T$.

Putting upper bounds for the first and second term together, we have $R_T \leq 4N\gamma_T LS + 4\sqrt{2}\left[(1/\sqrt{1-\epsilon}+1)\sqrt{\lambda}S + 2R\left(\sqrt{1+ND\beta}+N\sqrt{2D\beta}\right)\sqrt{\ln(1/\delta)+\gamma_T}\right]\sqrt{T\beta(1+N\beta D)\gamma_T}$.

*Proof of Lemma D.2.* We denote $\mathbf{V}_k = \lambda\mathbf{I} + \mathbf{\Phi}_{\mathcal{D}_k}^{\top}\mathbf{\Phi}_{\mathcal{D}_k}$, $\Delta\mathbf{V}_{k,s-1}(i) = \mathbf{\Phi}_{\mathcal{P}_{\underline{k}(i)}}^{\top}\mathbf{\Phi}_{\mathcal{P}_{\underline{k}(i)}}$ for $i \neq c_k$ and $\Delta\mathbf{V}_{k,s-1}(c_k) = \mathbf{\Phi}_{\{s' \in \mathcal{P}_k : s' \leq s-1\}}^{\top}\mathbf{\Phi}_{\{s' \in \mathcal{P}_k : s' \leq s-1\}}$.

In the following, we first show that

$$\mathbf{V}_k \succeq \frac{1}{\beta D}\Delta\mathbf{V}_{k,s-1}(i) \tag{10}$$

for all $i \in [N]$. Note that for any client $i \neq c_k$, we have

$$\frac{\mathbf{x}^{\top}\mathbf{V}_{\underline{k}(i)}^{-1}\mathbf{x}}{\mathbf{x}^{\top}\left(\mathbf{V}_{\underline{k}(i)} + \Delta\mathbf{V}_{k,s-1}(i)\right)^{-1}\mathbf{x}} \leq 1 + \sum_{s \in \mathcal{P}_{\underline{k}(i)}} \mathbf{x}_s^{\top}\mathbf{V}_{\underline{k}(i)}^{-1}\mathbf{x}_s$$

$$= 1 + \sum_{s \in \mathcal{P}_{\underline{k}(i)}} \sigma_{\underline{k}(i)}^2(\mathbf{x}_s) \leq 1 + \beta \sum_{s \in \mathcal{P}_{\underline{k}(i)}} \tilde{\sigma}_{\underline{k}(i)}^2(\mathbf{x}_s)$$

$$\leq 1 + \beta D,$$

where the first inequality is due to Lemma A.5, the second inequality is due to Lemma 4.1 and Lemma A.6, and the last inequality is due to equation 9.

This implies $\mathbf{V}_{\underline{k}(i)}^{-1} \preceq (1 + \beta D)\left(\mathbf{V}_{\underline{k}(i)} + \Delta\mathbf{V}_{k,s-1}(i)\right)^{-1}$. Then due to Lemma A.9, we have $(1 + \beta D)\mathbf{V}_{\underline{k}(i)} \succeq \mathbf{V}_{\underline{k}(i)} + \Delta\mathbf{V}_{k,s-1}(i)$, and thus $\mathbf{V}_{\underline{k}(i)} \succeq \frac{1}{\beta D}\Delta\mathbf{V}_{k,s-1}(i)$. Moreover, since $\underline{k}(i) < k, \forall i \neq c_k$, we have $\mathbf{V}_k \succeq \mathbf{V}_{\underline{k}(i)} \succeq \frac{1}{\beta D}\Delta\mathbf{V}_{k,s-1}(i)$. Similarly for client $c_k$, we have

$$\frac{\mathbf{x}^{\top}\mathbf{V}_k^{-1}\mathbf{x}}{\mathbf{x}^{\top}\left(\mathbf{V}_k + \Delta\mathbf{V}_{k,s-1}(c_k)\right)^{-1}\mathbf{x}} \leq 1 + \sum_{s' \in \mathcal{P}_k : s' \leq s-1} \sigma_k^2(\mathbf{x}_{s'}) \leq 1 + \beta D.$$

Again, this implies $\mathbf{V}_k \succeq \frac{1}{\beta D}\Delta\mathbf{V}_{k,s-1}(c_k)$, which finishes the proof of equation 10.

By averaging equation 10 over all $N$ clients, we have

$$\mathbf{V}_k \succeq \frac{1}{N\beta D}\sum_{i=1}^{N}\Delta\mathbf{V}_{k,s-1}(i),$$

and thus, we have

$$\mathbf{V}_k + \sum_{i=1}^{N}\Delta\mathbf{V}_{k,s-1}(i) \preceq (1 + N\beta D)\mathbf{V}_k.$$

Using Lemma A.9 again, we have $(1 + N\beta D)(\mathbf{V}_k + \sum_{i=1}^{N} \Delta\mathbf{V}_{k,s-1}(i))^{-1} \succeq \mathbf{V}_k^{-1}$. Therefore, we have

$$\frac{\sigma_k^2(\mathbf{x})}{\sigma_{k,s-1}^2(\mathbf{x})} \leq \frac{\phi(\mathbf{x})^\top \mathbf{V}_k^{-1} \phi(\mathbf{x})}{\phi(\mathbf{x})^\top (\mathbf{V}_k + \sum_{i=1}^{N} \Delta\mathbf{V}_{k,s-1}(i))^{-1} \phi(\mathbf{x})} \leq 1 + N\beta D$$

$\square$

