# OpenReview forum: "Learning Kernelized Contextual Bandits in a Distributed and Asynchronous Environment"
_ICLR.cc/2023/Conference — ICLR 2023 poster_

### Official Review · Reviewer_fUPj · 2022-10-24

**Confidence:** 2
**Clarity, Quality, Novelty And Reproducibility:** This paper is well-written and easy t…
**Correctness:** 3
**Technical Novelty And Significance:** 3
**Empirical Novelty And Significance:** 3
**Recommendation:** 8

**Strength And Weaknesses:**

Strength:

1. The author provided the first analyses of asynchronous kernelized contextual bandits and proposed a novel algorithm, Async-KernelUCB, with a theoretical guarantee.

2. Both the Synthetic and real-world experiments show that Async-KernelUCB outperforms other algorithms, which supports the efficiency of the Async-KernelUCB algorithm.

3. Compared with previous studies on kernelized contextual bandits, the Async-KernelUCB algorithm further solves the problem of different dictionary copies in an asynchronous environment, providing enough technique novelty.

Weakness:

In section 3.2, the author mentions that Nystrom approximation can help reduce computation and communication costs. However, it is better if the author directly indicates the computation complexity for equations (1) and (2) to show how Nystrom approximation improves efficiency.

**Summary Of The Paper:**

This work focused on kernelized contextual bandits with distributed computation and asynchronous communication. The author proposed the Async-KernelUCB algorithm with $\tilde O(\sqrt(T))$ regret and only $O(N^2)$ communication complexity. In addition, both the Synthetic and real-world experiments show that Async-KernelUCB outperforms other algorithms, which supports the efficiency of the Async-KernelUCB algorithm.

**Summary Of The Review:**

In summary, this paper provided the first analyses of asynchronous kernelized contextual bandits and proposed a novel algorithm, Async-KernelUCB with a theoretical guarantee, which makes a solid contribution to asynchronous kernelized contextual bandits. Unfortunately, due to the time limitation, I do not have enough time to check the proof in the appendix. Thus I choose a low confidence level.

---

> ### Author Response · Authors · 2022-11-12
> **Response to Reviewer fUPj**
>
> **[Q1]** “computation complexity for equations (1) and (2) to show how Nystrom approximation improves efficiency”
>
> **[R1]** We have updated the discussions in Section 3.2 to directly indicate the computation complexity as suggested by the reviewer.
>
> Specifically, the computation of Eq (1) involves a $|\mathcal{D}|$ dimensional vector $K_{\mathcal{D}}(x)$ and $|\mathcal{D}| \times |\mathcal{D}|$ matrix $K_{\mathcal{D}, \mathcal{D}}$, respectively, where $|\mathcal{D}|$ scales linearly with respect to time horizon $T$.
> Its computation complexity is $O(T^{3})$, due to taking the inverse of the matrix $K_{\mathcal{D}, \mathcal{D}}$. Moreover, raw data points $(x_{s},y_{s})$ for $s \in \mathcal{D}$ need to the transferred in order to compute $K_{\mathcal{D}}(x)$ and $K_{\mathcal{D}, \mathcal{D}}$, which leads to total communication cost that is linear in $T$.
>
> Therefore, instead of computing Eq (1), our algorithm computes Eq (2), which is an approximation of Eq (1) using Nystrom method. As mentioned on page 4, to compute Eq (2), the clients only need to transfer the $|\mathcal{S}|$ dimensional vector $Z_{\mathcal{D};\mathcal{S}}^{\top} y_{\mathcal{D}}$, and $|\mathcal{S}| \times |\mathcal{S}|$ matrix $Z_{\mathcal{D};\mathcal{S}}^{\top} Z_{\mathcal{D};\mathcal{S}}$, where $|\mathcal{S}|$ scales linearly with respect to the maximum information gain $\gamma_{T}$ and thus is generally much smaller than $|\mathcal{D}|$. This only incurs $O(\gamma_{T}^{2})$ communication cost, which is much smaller than that of Eq (1). Moreover, the computation complexity is also reduced from $O(T^{3})$ to $O(\gamma_{T}^{3})$.

---

### Official Review · Reviewer_iavx · 2022-10-24

**Confidence:** 3
**Clarity, Quality, Novelty And Reproducibility:** The paper is generally clear and ther…
**Correctness:** 3
**Technical Novelty And Significance:** 3
**Empirical Novelty And Significance:** 3
**Recommendation:** 6

**Strength And Weaknesses:**

Strength:
1. The work is well-organized and generally easy to follow. The motivation is clear.
1. One key among the algorithm design is that the communication threshold ensures that little rewards are not involved in the calculation and the communication cost is saved.
1. In Section 4.3, the author(s) clarifies that the key challenge --- to achieve low regret and low communication cost --- is overcome with a dictionary.
1. Experiment results look reasonable.

Weaknesses:
1. Is there a matching lower bound?

**Summary Of The Paper:**

This work is the first to learn kernelized contextual bandits in a distributed and asynchronous environment, which stems from the fact that it is hard to get feedback from all clients at the same time. It applies the idea of Nystrom approximation to help reduce communication cost.

**Summary Of The Review:**

This work focuses on the asynchronous setting. It claims to be the first to study this setting and provides both theoretical and numerical findings. However, except for the upper bound, it would be nice to provide a matching lower bound to convince the efficacy of the proposed Async-KernelUCB algorithm.


=============

After rebuttal:
Thanks for authors’ response. I would like to keep the score.

---

> ### Author Response · Authors · 2022-11-12
> **Response to Reviewer iavx**
>
> **[Q1]** “Is there a matching lower bound?”
>
> **[R1]** To facilitate our discussion about the communication lower bound, we should note that, for distributed bandit problems, the communication lower bound is only meaningful under different constraints/requirements on regret. The following two extreme cases of regret are of particular interest to explain the notion:
> *1)* run $N$ instances of an optimal bandit algorithm, e.g., LinUCB for linear bandit and KernelUCB for kernelized bandit, separately on each client with no communication, which leads to $O(\sqrt{NT})$ regret and $0$ communication cost;
> *2)* run one instance of the optimal bandit algorithm over the data of all $N$ clients, which leads to $O(\sqrt{T})$ regret and communication cost linear in $T$.
> The regret upper bound in case 2 is optimal, since it already matches the lower bound for a centralized bandit problem of time horizon $T$. Therefore, the goal of distributed bandit algorithms is to attain the optimal regret of $O(\sqrt{T})$, while having a communication cost sub-linear in $T$.
>
> The main contribution of our paper is to develop the first asynchronous algorithm that achieves such a goal for the distributed kernelized bandit problem. The lower bound analysis for the communication cost of federated bandits with respect to a specific regret requirement is highly non-trivial, and still remains an open problem. We leave this as an important future direction of ours.
>
> As mentioned in our conclusion, to the best of our knowledge, the only communication lower bound result available is for *case 1*. It was originally provided for the context-free setting (Theorem 2 of Wang et al. (2019)), but as shown in recent works, it also applies to the linear setting (Theorem 5.3 of He et al. (2022)), and kernelized setting with Square Exponential kernels (Theorem 19 of Li et al. (2022)). It states that, in order to have smaller regret than $O(\sqrt{NT})$ in *case 1*, an $\Omega(N)$ *number of communications* is necessary.
>
> So far, there is no useful result for *case 2*, i.e., the communication lower bound to obtain $O(\sqrt{T})$ regret; and none of the existing distributed bandit algorithms can close the gap with the $\Omega(N)$ lower bound for *case 1*. Specifically, to obtain $O(\sqrt{T})$ regret, DisLinUCB (Wang et al., 2019) requires $\tilde{O}(d^{3} N^{1.5})$ communication cost (requires additional assumption that clients appear in a round-robin manner),  AsyncLinUCB (Li and Wang, 2022) and FedLinUCB (He et al., 2022) require $\tilde{O}(d^{3} N^{2})$ communication cost, and Approx-DisKernelUCB (Li et al., 2022) requires $\tilde{O}(\gamma_{T}^{3} N^{2})$, with the latter three having the same scaling in $N$ as our proposed Async-KernelUCB.

---

### Official Review · Reviewer_6Kat · 2022-10-25

**Confidence:** 3
**Correctness:** 4
**Technical Novelty And Significance:** 2
**Empirical Novelty And Significance:** 3
**Recommendation:** 6

**Clarity, Quality, Novelty And Reproducibility:**

The paper is clearly written. My issue is with novelty as I'm not convinced that this presents a big enough improvement given the lack of motivation for the setting. Due to lack of time, I have not checked for reproducibility.

**Strength And Weaknesses:**

The paper presents a new setting and propose an algorithm that achieves near-optimal regret. Their empirical results are thorough and they have both synthetic data and real-world data.

My main weaknesses are:
1) If there is a new setting that is presenting, it needs to be backed up by stronger motivation as to where such a problem would occur
2) The algorithm contribution itself seems like a small improvement on current work when taking 1 into consideration

**Summary Of The Paper:**

The paper proposes a new setting of distributed kerneled contextual bandits: an asynchronous environment. For this setting, they propose an algorithm and show both theoretical and practical results. For the empirical results, they show the results on both synthetic and real-world data.

**Summary Of The Review:**

This paper that proposes a new setting and an algorithm to achieve near-optimal performance. The empirical results are well laid out. The main issues is with novelty and motivation for the asynchronous setting.

---

> ### Author Response · Authors · 2022-11-12
> **Response to Reviewer 6Kat**
>
> **[Q1]** “If there is a new setting that is presenting, it needs to be backed up by stronger motivation as to where such a problem would occur”
>
> **[R1]** We have added more discussions in Section 1 of our revised paper to better motivate the necessity of asynchronous communication as suggested by the reviewer. And here we would like to briefly summarize the newly added discussions.
>
> We would like to highlight that it has been widely recognized in existing literature in distributed/federated optimization [A1, A2, A3, A4, A5] and distributed bandit learning (Li and Wang, 2022, He et al., 2022) that, asynchronous approaches are more robust, when there are stragglers (i.e., slower workers) in the system. Due to device heterogeneity and network unreliability, this is often the case especially at the scale of hundreds or thousands of clients/devices. Take the synchronous solution for distributed kernelized bandit by Li et al. (2022) as an example. During each synchronization, the server needs to wait and collect the local dictionary and the embedded statistics from all the clients, and then send the aggregated update back. This approach is very sensitive to stragglers, as the whole update procedure is paused until the slowest client responds. In comparison, the asynchronous algorithm proposed in our paper is much more robust in this situation, as the server can readily perform model updates when the communication from a client is received, with no need to wait for other clients.
>
> **References**
> [A1] Xie, C., Koyejo, S. and Gupta, I., 2019. Asynchronous federated optimization. arXiv preprint arXiv:1903.03934.
> [A2] Lian, X., Zhang, W., Zhang, C. and Liu, J., 2018, July. Asynchronous decentralized parallel stochastic gradient descent. In International Conference on Machine Learning (pp. 3043-3052). PMLR.
> [A3] Chen, Y., Ning, Y., Slawski, M. and Rangwala, H., 2020, December. Asynchronous online federated learning for edge devices with non-iid data. In 2020 IEEE International Conference on Big Data (Big Data) (pp. 15-24). IEEE.
> [A4] Lim, W.Y.B., Luong, N.C., Hoang, D.T., Jiao, Y., Liang, Y.C., Yang, Q., Niyato, D. and Miao, C., 2020. Federated learning in mobile edge networks: A comprehensive survey. IEEE Communications Surveys & Tutorials, 22(3), pp.2031-2063.
> [A5] Low, Y., Gonzalez, J., Kyrola, A., Bickson, D., Guestrin, C. and Hellerstein, J.M., 2012. Distributed GraphLab: A Framework for Machine Learning and Data Mining in the Cloud. Proceedings of the VLDB Endowment, 5(8).
>
> **[Q2]** Improvement and novelty compared with prior work
>
> **[R2]** We want to emphasize that our paper is the first work in distributed bandit learning that enables *asynchronous communication beyond linear models*, i.e., the proposed solution simultaneously enjoys the modeling capacity of non-parametric models and the improved robustness against delays and unavailability of clients. This makes it suitable for a much wider range of applications, e.g., distributed recommender systems, or distributed Bayesian optimization for cyber-physical systems and automated machine learning, where communication bandwidth is a concern.
>
> And as mentioned in Section 1 of our paper, moving from synchronous communication to asynchronous communication causes several new challenges in both algorithmic design and theoretical analysis that cannot be addressed using the techniques of Li et al. (2022), and thus is highly non-trivial. Specifically, Li et al. (2022) required global synchronization because they adopted the standard Ridge leverage score (RLS) sampling procedure to update the dictionary as Calandriello et al. (2019, 2020), which requires sampling from the whole dataset using the same approximated variance function.
>
> To address this limitation, as illustrated in Figure 1 of our paper, we proposed a new variant of RLS sampling that allows each active client to incrementally update the dictionary and the embedded statistics stored on the server, *with no need to communicate with other clients*, as well as a linear transformation operation to convert the stored statistics received from different clients to a common subspace. To the best of our knowledge, our proposed asynchronous Nystrom approximation method, as well as its analysis (see discussions in Appendix B for details about technical novelties compared with prior works), is novel in bandits and machine learning literature in general, and it is also applicable to problems beyond bandit learning, e.g., distributed kernel online convex optimization.

---

### Official Review · Reviewer_yaEL · 2022-10-30

**Confidence:** 3
**Correctness:** 4
**Technical Novelty And Significance:** 3
**Empirical Novelty And Significance:** 4
**Recommendation:** 6

**Clarity, Quality, Novelty And Reproducibility:**

The claims in the paper are presented clearly, and the writing is good. The proposed algorithm seems novel due to its asynchronous feature. It is difficult to assess its superiority in theory, nevertheless, because the accepted upper bound on communication cost is still the same as the one from Li et al. (2022). Additionally, it could be better if the authors ran the tests over a larger number of trials and provided the variance.

**Details Of Ethics Concerns:**

I do not have any concerns.

**Strength And Weaknesses:**

Strength
* The newly proposed algorithm is asynchronous, allowing for larger practical applications since it does not require all clients to participate and wait for data transmission.
* The proposed algorithm is validated using both synthetic and real-world data.

Weaknesses
* Despite the author(s)' comprehensive trials demonstrating the newly proposed algorithm's superior communication cost compared to the synchrous one in Li et al. (2022), both of them have the same upper bound on regret and communication cost.
* The algorithms are only tested in three trials, and no variance information is given.

**Summary Of The Paper:**

This work studies distributed kernelized contextual bandits problem, where the reward function lies in a reproducing kernel Hilbert space. New asynchronous approach that does not require all clients to participate and wait for data exchange was proposed by the author(s). The suggested approach is shown to have near-optimal regret and sub-linear communication cost.

**Summary Of The Review:**

A new asynchronous approach for the distributed kernelized contextual bandits problem is proposed in this paper, which seems novel. Although it allows for more widespread application, the underlying premise is the same. Above all, I recommend weak accept.

---

> ### Author Response · Authors · 2022-11-12
> **Response to Reviewer yaEL**
>
> **[Q1]** “Despite the author(s)' comprehensive trials demonstrating the newly proposed algorithm's superior communication cost compared to the synchronous one in Li et al. (2022), both of them have the same upper bound on regret and communication cost.”
>
> **[R1]** As mentioned in Section 1, the main contribution of our work is to develop the first asynchronous algorithm for the distributed kernelized bandit problems. Compared with the synchronous algorithm by Li et al. (2022), our improvement focuses on the practical aspect, i.e., asynchronous algorithms are more robust to stragglers and communication delays, and thus preferred in distributed bandit problems (Li and Wang, 2022, He et al., 2022), and distributed/federated optimization in general [A1, A2, A3, A4, A5].
>
> Indeed, our proposed algorithm still has the same theoretical guarantee on regret and communication cost as its synchronous counterpart (Li et al., 2022). But we should emphasize that being able to recover the same theoretical guarantee as the synchronous solution is already non-trivial, due to the more stringent constraint on communication in the asynchronous setting. For example, for distributed linear bandits, the existing asynchronous solutions (Li and Wang, 2022, He et al., 2022) incur $O(\sqrt{N})$ more communication cost compared with their synchronous counterpart (Wang et al. 2019), in order to attain the optimal regret.
>
> **References**
> [A1] Xie, C., Koyejo, S. and Gupta, I., 2019. Asynchronous federated optimization. arXiv preprint arXiv:1903.03934.
> [A2] Lian, X., Zhang, W., Zhang, C. and Liu, J., 2018, July. Asynchronous decentralized parallel stochastic gradient descent. In International Conference on Machine Learning (pp. 3043-3052). PMLR.
> [A3] Chen, Y., Ning, Y., Slawski, M. and Rangwala, H., 2020, December. Asynchronous online federated learning for edge devices with non-iid data. In 2020 IEEE International Conference on Big Data (Big Data) (pp. 15-24). IEEE.
> [A4] Lim, W.Y.B., Luong, N.C., Hoang, D.T., Jiao, Y., Liang, Y.C., Yang, Q., Niyato, D. and Miao, C., 2020. Federated learning in mobile edge networks: A comprehensive survey. IEEE Communications Surveys & Tutorials, 22(3), pp.2031-2063.
> [A5] Low, Y., Gonzalez, J., Kyrola, A., Bickson, D., Guestrin, C. and Hellerstein, J.M., 2012. Distributed GraphLab: A Framework for Machine Learning and Data Mining in the Cloud. Proceedings of the VLDB Endowment, 5(8).
>
> **[Q2]** “The algorithms are only tested in three trials, and no variance information is given.”
>
> **[R2]** We have conducted additional experiments as suggested by the reviewer, and reported both mean and standard deviation of the results (over 10 trials) in the updated Figure 2 of the revised paper.
> With the newly added variance information, we observe that the regret of distributed linear bandit algorithms, DisLinUCB and FedLinUCB, have much larger variances (illustrated as shaded area in the updated Figure 2) across trials on most datasets. Again, this suggests linear models fail to capture the complicated reward mappings on these datasets, i.e., they tend to commit to pulling a particular arm early on, and therefore fail to identify the optimal policy.

---

### Author Response · Authors · 2022-11-17
**Common Response to All Reviewers**

Dear reviewers,

Thank you again for your valuable time and thoughtful comments. We have provided detailed responses, and additional experiment results in the revised paper to best answer the questions. As we are approaching the end of the discussion stage, we would like to know if your concerns have been addressed. We are more than happy to further discuss any details that you find not fully addressed. Thank you.

---

### Decision · Program_Chairs · 2023-01-20

**Decision:**

Accept: poster

**Justification For Why Not Higher Score:**

Due to the limited interest in the setting as well as the relatively small novelty compared to earlier work.

**Justification For Why Not Lower Score:**

Solid, publishable work with some novel ideas.

**Metareview: Summary, Strengths And Weaknesses:**

The paper studies kernelized distributed contextual bandits in the distributed setting. While earlier work in this topic required synchronization among the agents, here the authors provide a fully asynchronous learning algorithm achieving an optimal regret bound with the same communication complexity as the synchronous algorithm.

The weakness of the paper lies in the limited interest in the setting (kernelized contextual bandits) and the relatively limited novelty compared to the synchronous case.

**Note From Pc:**

if the above contains the word "oral" or "spotlight" please see: "oral" presentation means -> notable-top-5% and "spotlight" means -> notable-top-25%. As stated in our emails, we are disassociating presentation type from AC recommendations